# Understanding the Evolution of the Neural Tangent Kernel at the Edge of Stability

**Kaiqi Jiang**
Department of Electrical and Computer Engineering
Princeton University
Princeton, NJ 08540
kaiqij@princeton.edu

**Jeremy Cohen**
Machine Learning Department
Carnegie Mellon University
Pittsburgh, PA 15213
jeremycohen@cmu.edu

**Yuanzhi Li**
Machine Learning Department
Carnegie Mellon University
Pittsburgh, PA 15213
yuanzhil@andrew.cmu.edu

## Abstract

The study of Neural Tangent Kernels (NTKs) in deep learning has drawn increasing attention in recent years. NTKs typically actively change during training and are related to feature learning. In parallel, recent work on Gradient Descent (GD) has found a phenomenon called Edge of Stability (EoS), in which the largest eigenvalue of the NTK oscillates around a value inversely proportional to the step size. However, although follow-up works have explored the underlying mechanism of such eigenvalue behavior in depth, the understanding of the behavior of the NTK *eigenvectors* during EoS is still missing. This paper examines the dynamics of NTK eigenvectors during EoS in detail. Across different architectures, we observe that larger learning rates cause the leading eigenvectors of the final NTK, as well as the full NTK matrix, to have greater alignment with the training target. We then study the underlying mechanism of this phenomenon and provide a theoretical analysis for a two-layer linear network. Our study enhances the understanding of GD training dynamics in deep learning.

## 1 Introduction

Gradient Descent (GD) is a canonical minimization algorithm widely used in optimization and machine learning problems. While its behavior and guarantee in convex settings has been fully studied, there are still enormous challenges when understanding its dynamics in non-convex settings, especially for deep learning, where we usually encounter high-dimensional and highly non-convex losses. In recent years, the study on Neural Tangent Kernel (NTK) proposed in [JGH18] has been more and more popular when understanding the learning mechanism of GD, for example, [DZPS18] first proved the convergence of GD in the so-called *lazy* regime, where NTK remains unchanged during training. More and more follow-up studies point out the limitation of this lazy regime and advocate for the analysis in the *rich* regime where the NTK actively evolves during training. They argue that neural networks can learn more efficiently than kernel methods [GMMM20, WGL+20, DLS22]. These results motivate us to track the learning dynamics and the evolution of NTK more carefully.

There has also been a line of research on the behavior of NTK eigenvalues when training neural networks using GD. [CKL+21] observed a phenomenon called Edge of Stability (EoS) where the largest eigenvalue of loss Hessian first increases and then hovers around a value inversely proportional

39th Conference on Neural Information Processing Systems (NeurIPS 2025).

to the step size during training. This can also translate to similar behavior of the largest eigenvalue of NTK if we use MSE loss. There is a lot of follow-up work studying the underlying mechanism of this eigenvalue behavior (e.g. [WLL22, DNL23]). Despite the success of those works in studying the behavior of eigenvalues of NTK, the understanding of the rotation of its eigenvectors, especially those leading ones, is still lacking. Inspired by the evolution equation under Gradient Flow (GF),

$$\frac{d}{dt}(f_t - y) = -K_t(f_t - y)$$

where $y$ is the target vector, $f_t$ and $K_t$ represent the model output and NTK at time $t$, we are interested in studying the *alignment* between the NTK and the target, and how this alignment evolves over time.

The study of the alignment of NTK and the target is not a new story [KI20, SB21, WES$^+$23] and many different alignment metrics have been developed, e.g. Kernel Target Alignment (KTA) [CMR12].

**Definition 1.1** (Full Kernel Target Alignment (KTA)). *Let $y \in \mathbb{R}^m$ be the target vector and $K \in \mathbb{R}^{m \times m}$ be the NTK. Then we say that the* kernel target alignment *is* $\frac{y^T K y}{\|y\|_2^2 \|K\|_F}$.

However, there has been a lack of research on how the EoS interacts with the evolution of the NTK eigenvectors and the kernel target alignment. This motivates us to study the following question:

*How do eigenvectors of NTK evolve over time and align with the target, especially during EoS when eigenvalues of NTK have nontrivial oscillation?*

## 1.1 Overview of Our Main Contributions

This paper is dedicated to addressing the above question. Apart from the KTA in Definition 1.1, we are also interested in the *individual eigenvector target alignment* (see Definition 1.2). We aim to track the evolution of these two metrics during GD training and study their connection to the EoS.

**Definition 1.2** (Individual Eigenvector Target Alignment). *For the target vector $y \in \mathbb{R}^m$, if $u_k \in \mathbb{R}^m$ is the $k$-th eigenvector of the NTK, then we say that its* alignment *with the target is* $\frac{(y^T u_k)^2}{\|y\|_2^2}$. *Note that the alignment values sum up to 1, so they can be regarded as a kind of "distribution".*

We make the following contributions:

1. Across different architectures, we observe that larger GD learning rates cause the final NTK matrix to have higher KTA alignment (Definition 1.2) with the target, as shown in Figure 1a. Looking into the mechanism, we observe that larger learning rates cause the leading eigenvectors of the final NTK to have greater individual alignment with the target (Definition 1.2). Namely, larger learning rates cause the "distribution" of the Individual Eigenvector Target Alignment to *shift* towards leading eigenvectors, as shown in Figure 1b. We refer to this phenomenon as *alignment shift*.

2. We study the underlying mechanism of the alignment shift phenomenon by tracking the detailed training dynamics. We find that during the phases of EoS when the sharpness reduces, we usually observe sudden or fast increases in alignment, as shown in Figure 1c.

3. To better understand these empirical findings, we theoretically analyze the alignment shift phenomenon in a 2-layer linear network. This analysis reveals that the phase of EoS when the sharpness reduces is not only correlated with but essentially a contributing factor to the alignment shift. We also partially generalize our theoretical analysis to nonlinear networks in Appendix A.6.

4. To further link sharpness reduction with increased kernel-target alignment, we use the central flow framework [CDT$^+$24], which models gradient descent at EoS as implicitly following a sharpness-penalized gradient flow. We show empirically that this sharpness penalty leads to increased KTA.

## 2 Related Work

**Study on eigenvalues of NTK** [LBD$^+$20] showed that when wide neural networks are trained with a learning rate large enough to trigger initial instability, the largest eigenvalue of the NTK drops until stability is restored, an effect referred to as the "catapult mechanism." [CKL$^+$21] showed that when standard neural networks are trained using GD with a fixed learning rate, the largest Hessian eigenvalue (which is nearly equivalent to the largest NTK eigenvalue up to a constant under MSE

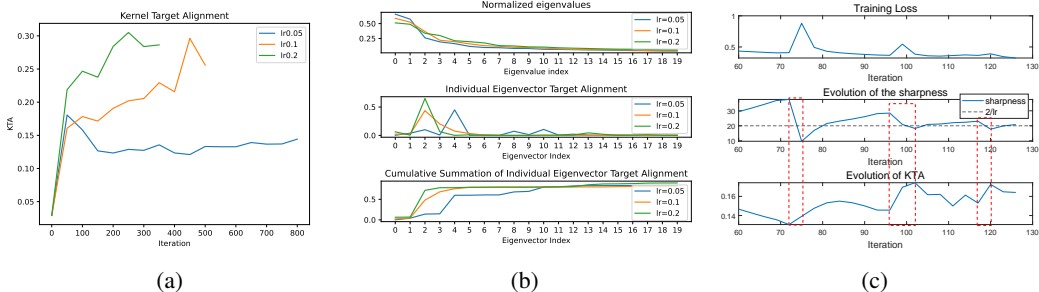

Figure 1: (a) KTA across learning rates, (b) the alignment shift, and (c) connection between the alignment dynamics and Edge of Stability in a fully-connected network (see Section 3.1). Note that (b) only plots the alignment on top 20 eigenvectors as later ones have much smaller alignment.

loss[WLL22]) rises until reaching the value $2/\eta$ (the threshold beyond which instability is triggered), and then oscillates or fluctuates around that value. This phenomenon was referred to as "Edge of Stability" (EoS). Later, [WLL22] analyzed the underlying mechanism of EoS in two-layer linear networks, and [DNL23] proposed a more general framework to understand its mechanism in more general settings. [GKW+25] studied the ranges of $\eta$ under which EoS oscillates with different ranks.

**Alignment between NTK and the target** Prior works have argued that the evolution of NTK and its alignment with the target is related to feature learning [WES+23]. [OJMDF21, SB21] observed that KTA is trending to increase during training. [KI20] found that top NTK eigenvectors tend to align with the learned target function, which improves the optimization. However, existing works either lack theoretical analysis of underlying mechanisms or focus on the GF setting or GD case before EoS, lacking study on the NTK behavior at EoS during which its eigenvalues have nontrivial oscillation.

**Relation between optimization dynamics and feature learning** [ZLRB23] argued that "catapult dynamics," which they defined as spikes in the training loss and decreases in the NTK maximum eigenvalue, promote better feature learning, as quantified by the alignment of the AGOP matrix [RBPB22] with the ground truth. The AGOP is an outer product matrix of the network gradients with respect to the *input*, whereas the NTK (studied here) is an outer product matrix of the network gradients with respect to the *weights*. On the two-layer linear network which we use in Section 3.1 and for theoretical analysis, we find that large learning rates and sharpness reduction are much more associated with enhanced NTK alignment than with enhanced AGOP alignment (see Appendix A.4). Note also that [ZLRB23] was limited to empirical observations and did not attempt to theoretically explain the mechanism by which catapult dynamics enhance AGOP alignment.

# 3 Empirical Observations about Alignment

## 3.1 Alignment Shift Phenomenon in Different Settings

**Linear networks** We start with a simple task on a two-layer linear network. We use Gaussian data $X$ as input where $X = Z\mathrm{diag}(3, 1.5, 1.2, 0.8, ..., 0.8)$ where each entry of $Z \in \mathbb{R}^{d \times n}$ is sampled i.i.d from the standard normal $\mathcal{N}(0, 1)$. Here, $n = 200$ is the number of training examples, and $d = 400$ is the input dimension. The target is a linear function of the input: $Y = \beta^T X$ with an all-one vector $\beta$. We use MSE loss and GD to train the model. Figure 2a plots the evolution of the KTA under different learning rates starting from the same initialization, showing that larger learning rates cause more significant increases of KTA. Figure 2b plots the alignment between $Y$ and top 5 NTK eigenvectors $(u_1, ..., u_5)$ for the final models, showing that large learning rates cause alignment shift towards leading eigenvectors, while do not substantially change the normalized eigenvalue distribution (where the normalization is by the $l_2$-norm of the eigenvalue list). Figure 2c plots the dynamics of the training loss, sharpness, and individual eigenvector target alignment under a learning rate 0.014, showing that the alignment shift occurs during the period when the sharpness decreases.

**Fully-connected networks** We trained a 4-layer GeLU network using GD under MSE loss on a subset of CIFAR-10 [KH+09] where we randomly selected about 2000 examples in two classes (roughly 1000 for each class) and relabeled the target values as $\pm 1$. We trained the model from the

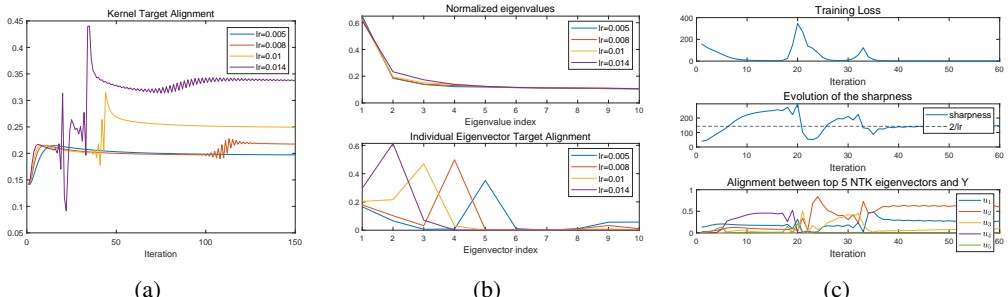

Figure 2: (a) KTA across learning rates, (b) the alignment shift, and (c) connection between the alignment dynamics and Edge of Stability in a 2-layer linear network (see Section 3.1).

same initialization under different learning rates to roughly the same training loss. We plotted the evolution of KTA (every 50 steps) in Figure 1a and the individual eigenvector target alignment for the final models in Figure 1b. As is mentioned in Section 1, we see the increase of KTA and the alignment shift. Figure 1c zooms in the EoS period and plots the evolution of the sharpness and KTA (every 3 steps). We see that during the periods when the sharpness decreases, the KTA usually has a sudden or fast increase. During the periods when the sharpness increases, the KTA increases slower or can decrease, i.e. the "speed" of increase is lower. Table 1a provides a quantitative analysis of the above connection, where we compute the total and average change (per step) of KTA during each sharpness-decreasing period and sharpness-increasing period. We see that the average change in KTA per step, which can be regarded as the average changing "speed" of KTA, is positive and much larger in the sharpness-decreasing period than in the sharpness-increasing period. In Section 5, we study the above connection from another perspective using a tool called *central flow* developed by [CDT$^+$24] and demonstrate that adding a sharpness penalty to gradient flow can promote the increase of KTA.

**Vision Transformer** On the same dataset as above, we also trained a simple ViT[1] using GD under MSE loss from the same initialization under different learning rates. Figure 3a plots the evolution of KTA (every 20 iterations) when training to roughly the same loss. Figure 3b shows the individual eigenvector target alignment as well as its cumulative summation for the final models. Again, we see the increase of KTA and the alignment shift phenomenon. Figure 3c zooms in the EoS period and plots the evolution of sharpness and the evolution of KTA every 5 steps. Here, we see a stair-wise behavior of KTA where, after each period when the sharpness decreases, the KTA value goes up to the next "stair". We also add numerical evidence in Table 1b to make a quantitative analysis, where we again see the connection between the decrease of sharpness and the fast increase of KTA per step.

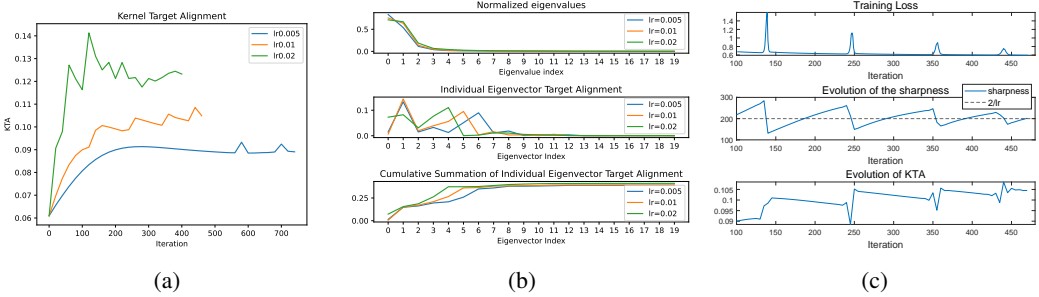

Figure 3: (a) KTA across learning rates, (b) the alignment shift, and (c) connection between the alignment dynamics and Edge of Stability in a simple ViT (see Section 3.1) Similar to Figure 1b, we plot the alignment of the first 20 eigenvectors in (b).

## 3.2 Summary and Discussion

We demonstrate that larger learning rates cause the NTK matrix as well as its leading eigenvectors to have higher alignment with the targets, and reveal the connection between this alignment behavior and EoS. In Appendix A.2, we provide supplementary results to demonstrate that our observations also

---

[1] `https://github.com/lucidrains/vit-pytorch`. MIT license.

Table 1: Comparison of the changing speed of KTA between the sharpness-decreasing periods and sharpness-increasing periods on (a) fully-connected network and (b) ViT. See Section 3.1 for details.

(a)

| Iteration interval | 72-75 | 75-96 | 96-102 | 102-117 | 117-120 |
|---|---|---|---|---|---|
| Sharpness behavior | decrease | increase | decrease | increase | decrease |
| Total change of KTA | 0.0086 | 0.0065 | 0.0287 | -0.021 | 0.01948 |
| Avg change of KTA per step | 0.0029 | 0.0003 | 0.0048 | -0.0014 | 0.0065 |

(b)

| Iteration interval | 130-145 | 145-235 | 235-255 | 255-345 | 345-365 | 365-425 |
|---|---|---|---|---|---|---|
| Sharpness behavior | decrease | increase | decrease | increase | decrease | increase |
| Total change of KTA | 0.0100 | -0.0034 | 0.0065 | -0.0042 | 0.0046 | -0.0024 |
| Avg change of KTA per step | 6.68e-4 | -3.83e-5 | 3.26e-4 | -4.67e-5 | 2.30e-4 | -3.99e-5 |

hold in many other settings, such as different architectures, multi-class tasks, language transformers. Although we focus on GD for simplicity, in Appendix A.3, we show that our observations also generalize to the SGD setting. Now we add more discussion on the implication of our findings.

**Connection to feature learning**. Our work provides an interesting perspective on the beneficial effects of large learning rates on deep learning. While a common perspective is that large learning rates find "flatter minima" which generalize better [HS97], our findings support the (potentially complementary) view that large learning rates improve feature learning [AVPVF23, ZLRB23], in our case by enhancing the alignment between the NTK and the target vector.

**Connection to the generalization ability**. [ADH$^+$19] proposed a generalization bound in terms of the alignment between inverse NTK $K^{-1}$ and the target $y$, which is proportional to $\sqrt{\frac{y^T K^{-1} y}{n}}$ where $n$ is the number of training examples. Hence, if $y$ gets more alignment with earlier eigenvectors of $K$, i.e. later eigenvectors of $K^{-1}$, we will expect to have a smaller value of $\sqrt{\frac{y^T K^{-1} y}{n}}$ and a better generalization. This fact, together with our alignment shift observation, can help us understand the generalization ability of large learning rates. Appendix A.5 provides empirical results to demonstrate that the reducing sharpness during EoS and the alignment shift towards earlier eigenvectors are correlated with a sudden decrease of the normalized $y^T K^{-1} y$, and that larger learning rates lead to more significant decreases. We also directly examine the change of the test loss over time and observe the improvement of the generalization ability after the zigzag period when alignment shift occurs.

**Results on overlarge learning rates.** Finally, we want to point out that the correlation between better generalization ability and better kernel target alignment discussed above also holds for overlarge learning rates. Table 2 shows the change of the kernel target alignment under several large learning rates. We observe that 1) an overlarge learning rate may lead to bad test performance, 2) in the meantime, the alignment gets worse, too. That means our focus on the kernel target alignment is a reasonable choice to understand the effect of the learning rate on the generalization ability.

Table 2: Generalization ability and KTA on large learning rates. Experiments were conducted in the fully-connected setting using SGD with batch size 256. All models were trained to the same training loss of 0.01.

| Learning rates | 0.02 | 0.025 | 0.03 | 0.035 |
|---|---|---|---|---|
| Test losses | 0.4290 | 0.4244 | 0.4241 | 0.4664 |
| Kernel Target Alignment (KTA) | 0.3607 | 0.3669 | 0.3878 | 0.3432 |

## 4 Theoretical Analysis on Two-layer Linear Networks

To better understand the mechanism of the empirical results presented in Section 3, in this section we provide a theoretical analysis of the alignment behavior during EoS on a two-layer linear network.

### 4.1 Setup

**Notation** We use $\|\cdot\|_2$ to denote the $l_2$ norm of a vector, and $\|\cdot\|_F$ to denote the Frobenius norm of a matrix. Let $\langle \cdot, \cdot \rangle$ be the inner product between vectors or matrices. Let $X[i]$ (resp. $X[i,j]$) denote the $i$-th (resp. $(i,j)$-th) component of a vector (resp. matrix) $X$. For a quantity $A$ which evolves throughout training, we use $A_t$ to denote its value at iteration $t$.

**Model setup**   Our training dataset consists of $n$ datapoints in $d$ dimensions, represented by the data matrix $X \in \mathbb{R}^{d \times n}$ and the label matrix $Y \in \mathbb{R}^{1 \times n}$. We will use $\{q_i\}$ and $\{\lambda_i\}$ ($\lambda_1 \geq \lambda_2 \geq ... \geq \lambda_n$) to denote the eigenvectors and eigenvalues, respectively, of the data kernel $K_x := X^T X$. We also use $\{p_i\}$ to denote the eigenvectors of the covariance matrix $XX^T$.

We consider a two-layer linear network with one-dimensional output which maps $\mathbb{R}^d \to \mathbb{R}$: $x \mapsto W^{(2)}W^{(1)}x$ with parameters $W := (W^{(2)}, W^{(1)})$, where $W^{(1)} \in \mathbb{R}^{d_h \times d}$ and $W^{(2)} \in \mathbb{R}^{1 \times d_h}$ and $d_h$ is the hidden layer size. We train the network using GD with the square loss function $L(W) := \frac{1}{2n} \left\| W^{(2)}W^{(1)}X - Y \right\|_F^2$. By simple calculation, the NTK at iteration $t$ is given by:

$$K_t = \left\| W_t^{(2)} \right\|^2 X^T X + X^T W_t^{(1)T} W_t^{(1)} X \tag{1}$$

**Data distribution**   Many prior works (e.g. [AGCH19, ABP22]) analyzing GD in linear networks use whitened data. On the other hand, most EoS analyses assume that the spectrum of the NTK/Hessian has just one outlier eigenvalue, as this setting results in only one direction of oscillation (which is more tractable to analyze). This typically requires the data kernel $X^T X$ to have one or several outlier eigenvalues, e.g. assumed in [WLL22]. Therefore in our case, we use the following data with several unwhitened leading eigenvalues, but a roughly whitened tail.

Assume that $\lambda_1 = \Theta(n)$. Suppose there exists $k$ such that 1) for the first $k$ eigenvalues, $\forall i \leq k$, $\lambda_i \leq \lambda_{i-1}/\gamma$ with some $\gamma > 1$; 2) the later $n - k$ eigenvalues have the order $\Theta(n^a)$ with $a \in (0, 1)$, and that $\lambda_k - \frac{1}{n-k} \sum_{i=k+1}^{n} \lambda_i = \delta_\lambda n^a$ with $\delta_\lambda = o(1)$. We use a linear function of our input as the target, i.e. $Y = \beta^T X$ with $\|Y\|_2 = \Theta(\sqrt{n})$. Further, recalling that $\{p_i\}$ denotes the eigenvectors of $XX^T$, we assume $\forall i \leq n : \left| \left\langle \frac{\beta}{\|\beta\|}, p_i \right\rangle \right| = \Theta\left(\frac{1}{\sqrt{n}}\right)$. This is to assume that $\beta$ is evenly distributed across the principal components of the data, reflecting the intuition that $\beta$ is independent of $X$.

**Rank-1 structure of $W_t^{(1)}$**   Existing papers have shown that weight matrices approach low ranks during training [GWB+17, LMZ18, CGMR20]. For example, [ABP22, JML24] focus on whitened data (i.e. $\frac{1}{n}XX^T$ is identity) and argued that if the initialization is small, then the weights will have the following approximate updates since $W_2 W_1 X$ is small during early training iterations:

$$W_{t+1}^{(1)} \approx W_t^{(1)} + \eta/n \cdot W_t^{(2)T} \Lambda_{yx}, \quad W_{t+1}^{(2)} \approx W_t^{(2)} + \eta/n \cdot \Lambda_{yx} W_t^{(1)T}$$

where $\eta$ is the learning rate and $\Lambda_{yx} = YX^T$. Based on these approximate updates, [JML24] proved that under small initialization, the weights have the approximate rank-1 structure $W_{t_0}^{(1)} \approx u_{t_0} z_{t_0}^T$, $W_{t_0}^{(2)} \approx c_{t_0} u_{t_0}^T$ with some $c_{t_0} > 0$ after a short time $t_0$. They also proved that after time $t_0$, this approximate structure is preserved and that the $u_t$ remains unchanged and $c_t, v_t$ are quantities which actively change. This allows us to write $W_t^{(1)}, W_t^{(2)}$ for $t > t_0$ as $W_t^{(1)} \approx u z_t^T, W_t^{(2)} \approx c_t u^T$.

Our paper focuses on the general case with unwhitened $XX^T$. Since eq. (1) involves $W_t^{(1)}X$, we now focus on the update of $W_t^{(1)}X$ and $W_t^{(2)}$ and apply a similar argument as above. Then we have

$$W_{t+1}^{(1)}X \approx W_t^{(1)}X + \eta/n \cdot W_t^{(2)T} Y XX^T, \quad W_{t+1}^{(2)} \approx W_t^{(2)} + \eta/n \cdot Y(W_t^{(1)}X)^T$$

We can use a similar argument as in [ABP22, JML24] to show that $W_t^{(1)}X$ and $W_t^{(2)}$ quickly converge to the approximate structure $W_{t_0}^{(1)}X \approx u v_{t_0}^T$, $W_{t_0}^{(2)} \approx c_{t_0} u^T$ after a short time $t_0$ with $v_{t_0} = X^T z_{t_0}$. In the following analysis, we will assume this approximate structure is exact.

**Assumption 4.1.** *Assume that at some early iteration $t_0$, we have $W_{t_0}^{(1)}X = u v_{t_0}^T, W_{t_0}^{(2)} = c_{t_0} u^T$, with $c_{t_0} \in \mathbb{R}, u \in \mathbb{R}^{d_h}, v_{t_0} \in \mathbb{R}^n$. Without loss of generality, assume that $c_{t_0} > 0$ and $\|u\|_2 = 1$. Also assume small network output at $t_0$ such that $\forall i \leq n, |F_{t_0}[i]| = o(Y[i])$.*[2]

In Appendix B.2 we prove that if this rank-1 structure holds at time $t_0$, then it will be preserved for $t > t_0$. This allows us to focus on the dynamics of $c_t$ and $v_t$. In particular, the network output $F_t$ and the NTK $K_t$ can be written as $F_t = W_t^{(2)} W_t^{(1)} X = c_t v_t, K_t = c_t^2 X^T X + v_t v_t^T$. We also define the error vector $E_t := F_t - Y$. Here and throughout, we use standard notations $\mathcal{O}(\cdot), \Omega(\cdot), \Theta()$ and $o(\cdot)$ to represent order of quantities when $n \to \infty$.

---

[2]This assumes that $t_0$ is in early training, which has been proven or verified before, e.g. in [ABP22, JML24].

## 4.2 Four Phases of the Training Dynamics

[WLL22, DNL23] revealed that when there is one eigenvalue at EoS, the training dynamics can be divided into four phases based on the position and the sign of change in the sharpness (see Appendix B.1 for a cartoon illustration). In the two-layer linear network setting, [WLL22] demonstrated that the largest eigenvalue of the Hessian is nearly equivalent to the largest eigenvalue of the NTK, and that this is in turn nearly equivalent to the norm squared of the second-layer weight matrix times the largest eigenvalue of the data kernel: $\lambda_{\max}(\nabla^2 L(W_t)) \approx \frac{1}{n}\lambda_{\max}(K_t) \approx \frac{\lambda_1}{n}c_t^2$. Since $\lambda_1$ is constant, the four phases can therefore be specified by the position and the sign of change in $c_t^2$.

While [WLL22] essentially ignored the quadratic term of the step size, i.e. $\eta^2$, our results reveal that $\eta^2$ plays a crucial role on the alignment shift. More details can be found later in Phase III analysis.

**Phase I: $\frac{\lambda_1}{n}c_t^2 < \frac{2}{\eta}$ and is increasing**  During this phase, the training dynamics are stable and approximately follow the GF trajectory. Under GF updates, we have that

$$\frac{d}{dt}c_t^2 = -\frac{2\eta}{n}\langle E_t, F_t\rangle = -\frac{2\eta}{n}\sum_{i=1}^{n}\langle E_t, q_i\rangle\langle F_t, q_i\rangle, \quad \frac{d}{dt}\|\boldsymbol{v}_t\|^2 = -\frac{2\eta}{n}\sum_{i=1}^{n}\lambda_i\langle E_t, q_i\rangle\langle F_t, q_i\rangle \quad (2)$$

where $q_i$ is the $i$-th eigenvector of the data kernel $X^T X$ mentioned before. In Appendix B.3 we prove that under GF, $\forall i \leq n : \langle E_t, q_i\rangle\langle F_t, q_i\rangle < 0$ for $t > t'$ with some $t'$, making $\langle E_t, F_t\rangle < 0$ and $c_t^2$ increase. If the learning rate is very small, the loss will converge before the sharpness increases to the stability threshold $2/\eta$. Otherwise, the sharpness will increase to $2/\eta$ during training, causing the dynamics to go unstable and move to Phase II discussed below. To ensure that this occurs, we need to carefully pick our learning rate. Appendix B.4 discusses the intuition on the choice of $\eta$ in detail, where we pick $\eta = \Theta(n^{-\frac{1-a}{2}})$. Noticing the $\|\boldsymbol{v}_t\|^2$ update in eq. (2), we also have that $\|\boldsymbol{v}_t\|^2$ increases as well during Phase I. A more careful analysis can be found in Appendix B.3.

Here we see the behavior of $c_t^2$ and $\|\boldsymbol{v}_t\|^2$ is consistent, in that they are both increasing. This will not always be true in other phases, especially in Phase III. It turns out that the different behavior of $c_t^2$ and $\|\boldsymbol{v}_t\|^2$ in Phase III is the key contributing factor to the alignment shift phenomenon. For later phases, we make the following assumptions.

**Assumption 4.2.** *Assume that during the whole training procedure, $\frac{\lambda_1}{n}c_t^2 \leq \frac{C}{\eta}$ for some constant $C > 0$ and that $\frac{\lambda_i}{n}c_t^2 < \frac{1}{\eta}$ for $i \geq 2$.*

The first part of Assumption 4.2 stipulates that the largest eigenvalue of the approximate NTK is smaller than $\frac{C}{\eta}$, an assumption previously made in [WLL22] with $C = 4$. The second part stipulates that other eigenvalues are small, which was also assumed in [WLL22].

**Assumption 4.3.** *Assume that during the training procedure, $\|E_t\|^2 \leq \mathcal{O}(n/\eta)$. Also assume that at the end of Phase I, the loss has undergone a nontrivial decrease: $\|E_t\| = \delta_0\|Y\|$ with $\delta_0 = o(1)$. Further denote $\delta_1 := |\cos(Y, q_1)| = |\langle Y, q_1\rangle|/\|Y\|$, $\delta_2 := \max\{\delta_0, \sqrt{\delta_1}\}$.*

The first part of Assumption 4.3 assumes that the loss does not diverge too much during training. Because we pick $\eta = \Theta(\frac{1}{\text{poly}(n)})$ mentioned before and the initial $\|E_0\|^2$ is at the order $\mathcal{O}(\|Y\|^2) = \mathcal{O}(n)$, the assumption that $\|E_t\|^2 \leq \mathcal{O}(n/\eta)$ is saying that the loss is no larger than poly$(n)$ times the initial loss, which is not a strong assumption.

**Phase II: $\frac{\lambda_1}{n}c_t^2 > \frac{2}{\eta}$ and is still increasing.**  During this phase, the approximate sharpness has risen above $\frac{2}{\eta}$, which will lead to the oscillation of the network output along some direction. More precisely, we prove in Lemma 4.4 that the error vector $E_t$ will oscillate with increasing magnitude along the $q_1$ direction, whereas $\langle E_t, q_i\rangle$ for $i \geq 2$ will remain small in magnitude.

**Lemma 4.4.** *Under our data distribution (Section 4.1) and Assumption 4.1-4.3. Pick $\eta = \Theta(n^{-\frac{1-a}{2}})$ as discussed before. Write $\Delta_t := \frac{\lambda_1}{n}c_t^2 - \frac{2}{\eta}$ with $\Delta_t > 0$. Then for $t$ during Phase II, we have that*

$$|\langle E_t, q_1\rangle| \geq (1 + \eta\Delta_{t-1})|\langle E_{t-1}, q_1\rangle| - \mathcal{O}(\eta^2\delta_2|\langle Y, q_1\rangle|), \quad \sum_{i=2}^{n}\langle E_t, q_i\rangle^2 \leq \mathcal{O}\left(\delta_2^2\|Y\|^2\right) = o(\|Y\|^2)$$

Recall from Phase I that under the gradient flow, the time derivative of $c_t^2$ was determined by the sign of $\langle E_t, F_t \rangle$. We prove in Appendix B.5 that in Phase II and later phases, even though GD no longer follows the gradient flow, the sign of $c_{t+1}^2 - c_t^2$ is still determined by the sign of $\langle E_t, F_t \rangle$. That means in Phase II, $c_t^2$ will keep increasing as long as $\langle E_t, F_t \rangle < 0$. However, since Lemma 4.4 shows the divergence of $E_t$ along the $q_1$ direction, once $|\langle E_t, q_1 \rangle|$ grows sufficiently large, we have $\|E_t\| > \|Y\|$, which yields $\langle E_t, F_t \rangle = \|E_t\|^2 + \langle E_t, Y \rangle \geq \|E_t\|^2 - \|E_t\|\|Y\| > 0$, causing $c_{t+1}^2 < c_t^2$ and we move to Phase III. The detailed analysis of Phase II is presented in Appendix B.6.

**Phase III:** $\frac{\lambda_1}{n} c_t^2 > \frac{2}{\eta}$ **and starts to decrease.** As discussed in Phase II, the divergence of $E_t$ along the $q_1$ direction eventually causes $c_t^2$ to decrease. In Phase III, $c_t^2$ will keep decreasing until the sharpness (which is approximately $\frac{\lambda_1}{n} c_t^2$ in our case) goes below $\frac{2}{\eta}$ and $|\langle E_t, q_1 \rangle|$ starts to shrink. Then we will move to Phase IV. As for $\|v_t\|^2$, we can prove that $\|v_t\|^2$ is *increasing* in Phase III. In particular, the one-step evolution of $\|v_t\|^2$ is given by the following lemma.

**Lemma 4.5.** *We have that*

$$\|v_{t+1}\|^2 - \|v_t\|^2 = \frac{2\eta + \Delta_t \eta^2}{n} \cdot \sum_{i=2}^{n} \frac{\lambda_i^2}{\lambda_1} \langle E_t, q_i \rangle^2 + \frac{\Delta_t \eta^2}{n} \lambda_1 \langle E_t, q_1 \rangle^2 - \frac{2\eta}{n} \sum_{i=2}^{n} \lambda_i \langle E_t, q_i \rangle^2 - \frac{2\eta}{n} E_t K_x Y^T$$

*where $\Delta_t$ is defined in Lemma 4.4.*

By the divergence of $|\langle E_t, q_1 \rangle|$, we can prove that the RHS is dominated by the second term $\frac{\Delta_t \eta^2}{n} \lambda_1 \langle E_t, q_1 \rangle^2$ (see Appendix B.7), which does not exist under GF update and was usually ignored in prior works. Note that in Phase III, $\frac{\lambda_1}{n} c_t^2 > \frac{2}{\eta}$, and hence $\Delta_t > 0$. This leads to the increase of $\|v_t\|^2$. In the following section, we will see that this increasing behavior of $\|v_t\|^2$, which is different from the decrease of $c_t^2$, is the key contributing factor to the alignment shift, suggesting that the typically ignored $\eta^2$ term actually plays a crucial role on the alignment behavior during EoS.

**Phase IV:** $\frac{\lambda_1}{n} c_t^2 < \frac{2}{\eta}$ **and is still decreasing.** In Phase IV, $E_t$ starts to shrink along the $q_1$ direction. [WLL22] showed that we eventually have $\langle E_t, F_t \rangle < 0$ and go back to Phase I. Our paper will not analyze Phase IV in detail but will study the *overall* behavior after going through Phase IV.

### 4.3 Alignment Shift During the Sharpness Reduction Phases (III and IV)

Now we are ready to analyze the alignment shift phenomenon. We begin by defining the following ratio between $c_t^2$ and $\|v_t\|^2$, which turns out to be a crucial quantity when analyzing alignment shift.

**Definition 4.6.** *Define $\alpha_t := \frac{\|v_t\|^2}{c_t^2}$.*

We note that the sharpness reduction phases (III and IV) are typically correlated with the spikes of the loss (as is shown empirically in Section 3.1). [CKL+21, WLL22] have observed a non-monotonically decreasing behavior of the loss where it has a decreasing trend in spite of oscillation. In other words, the loss after the spike is typically around or smaller than the value before the spike. This motivates us to introduce the following assumption on the change of loss after going through phases III and IV.

**Assumption 4.7.** *Let $t_1$ be the start of Phase III and $t_2$ the end of Phase IV. Assume $\|E_{t_2}\|^2 \leq \|E_{t_1}\|^2$.*

**Theorem 4.8.** *Denote $\Delta_t := \frac{\lambda_1}{n} c_t^2 - \frac{2}{\eta}$. Under our data distribution (see Section 4.1) and Assumption 4.1-4.3 and pick $\eta = \Theta(n^{-\frac{1-a}{2}})$ as discussed before.*

*(A). For $t$ in Phase III such that $|\langle E_t, q_1 \rangle| \geq \Omega(\delta_2 \|Y\|)$ with $\delta_2$ defined in Lemma 4.4 and $\Delta_t > \Omega(\max\{1, \frac{1}{\eta n^{a/4}}\})$, we have that $c_{t+1}^2 < c_t^2$ and $\|v_{t+1}\|^2 > \|v_t\|^2$.*

*(B). Suppose we further have Assumption 4.7. For $t_1, t_2$ defined in Assumption 4.7, if $\Delta_{t_1} \geq \Omega(\frac{\delta_2}{\eta})$, then we will get that $\alpha_{t_2} > \alpha_{t_1}$ and that $cos(v_{t_1}, Y) \geq 1 - \mathcal{O}(\delta_2), cos(v_{t_2}, Y) \geq 1 - \mathcal{O}(\delta_2)$.*

Theorem 4.8 (A) implies a one-step increasing behavior of $\alpha_t$ in Phase III. In Phase IV, although $\alpha_t$ may not have this one-step increase, Theorem 4.8 (B) shows that it will have an *overall* increase after going through the whole Phase III and IV, i.e. $\alpha_{t_2} > \alpha_{t_1}$. It also shows that at times $t_1$ and $t_2$, $v_t$ approximately aligns with the training target $Y$, which allows us to write $K_t$ at $t_1$ and $t_2$ as $K_t \approx c_t^2 X^T X + \|v_t\|^2 \hat{y}\hat{y}^T$ where $\hat{y} := Y^T/\|Y\| \in \mathbb{R}^n$ is the unit vector.

We now argue that in the above approximate structure at $t_1$ and $t_2$, a larger $\alpha_t$ causes $\hat{y}$ to align with earlier NTK eigenvectors, as opposed to the later ones. Intuitively, if $\|v\|^2$ is large relative to $c^2$ (i.e. if $\alpha = \|v\|^2/c^2$ is large), then early eigenvectors of $K$ will align with $\hat{y}$, rather than with the top eigenvectors of $X^T X$. The following lemma gives a sufficient condition (involving $\alpha$ and the eigenvalue spectrum of $X^T X$) for the $j$-th NTK eigenvector to be the one with the highest alignment with $\hat{y}$. The lemma is just a property of eigenvalue spectra, and does not involve training dynamics. Hence for ease of notation, we drop the iteration subscript $t$ in the lemma statement. We provide an illustration of it in a warm-up setting in Appendix C.1 and proofs in the general case in Appendix C.2

**Lemma 4.9.** *Let $\tilde{q}_1, ..., \tilde{q}_n$ be the eigenvectors of the approximate NTK matrix $c^2 X^T X + \|v\|^2 \hat{y}\hat{y}^T$ with corresponding non-increasing eigenvalues, and let $\alpha := \frac{\|v\|^2}{c^2}$. Consider the $k$ and $\delta_\lambda$ defined in our data distribution in Section 4.1. For any $j \leq k-1$, if $\alpha$ satisfies $\frac{\lambda_j - \lambda_k}{1 - \mathcal{O}(\delta_\lambda + n^{-a/2})} < \alpha < \frac{\lambda_{j-1} - \lambda_k}{1 + \mathcal{O}(\delta_\lambda + n^{-a/2})}$, we have that $\arg\max_{1 \leq i \leq n} |\langle \tilde{q}_i, \hat{y} \rangle| = j$.*

Theorem 4.8 (B) and Lemma 4.9 suggest that $Y$ tends to align more with earlier NTK eigenvectors at $t_2$ than at $t_1$. This is the alignment shift trend after sharpness reduction periods, i.e. Phase III and IV.

The above discussion reveals the alignment shift trend in a single sharpness reduction period. We highlight that for the long-term behavior after many oscillation phases, this trend still holds. To see this, note that during EoS, $c_t^2$ oscillates around a fixed value $\frac{2n}{\eta\lambda_1}$. As for $\|v_t\|^2$, the analysis in Appendix B.3 proves its increasing behavior in Phase I. By proof details in Appendix B.7, we know that the behavior of $\|v_t\|^2$ in Theorem 4.8 (A) also applies to Phase II, which gives the increasing trend of $\|v_t\|^2$ during Phase II and III. During Phase IV it may decrease for some steps but will have an overall increase after going through Phase III and IV, according to Theorem 4.8 (B). Then we know that the long-term trend of $\|v_t\|^2$ is to increase, which gives us a long-term increasing trend of $\alpha_t$ and thus a long-term tendency to increase the alignment between $Y$ and earlier NTK eigenvectors.

In Appendix A.6 we generalize the above crucial quantity $\alpha_t$ in Definition 4.6 to nonlinear networks.

## 5   Leveraging Central Flows

In this section, we present further evidence that sharpness reduction at EoS improves the kernel target alignment. We leverage the recent central flows technique [CDT$^+$24], which models the time-averaged (i.e. locally smoothed) GD trajectory using an ODE called a *central flow* of the form:

$$\frac{dW}{dt} = -\eta \left[ \nabla L(W) + \underbrace{\nabla \langle \Sigma(t), \nabla^2 L(W) \rangle}_{\text{sharpness penalty}} \right]. \tag{3}$$

Here, $\Sigma(t)$ is a particular matrix defined as the solution to a convex optimization problem, and $\langle \cdot, \cdot \rangle$ denotes the Frobenius inner product between two matrices (i.e. the dot product of the two matrices when they are flattened into vectors), so the quantity $\langle \Sigma(t), \nabla^2 L(W) \rangle$ is a metric of sharpness where each entry of the Hessian $\nabla^2 L(W)$ is weighted by the corresponding entry of $\Sigma(t)$. The central flow eq. (3) penalizes this quantity and is therefore a *sharpness-penalized gradient flow*. [CDT$^+$24] shows that central flow averages out the oscillations while retaining their effect on the time-averaged gradient descent trajectory, which takes the form of this sharpness penalty. They demonstrate that central flow is a good approximation to the time-averaged GD. See more evidence in Appendix A.7.

In Figure 4a, we run both GD and the central flow at different learning rates in the fully-connected setting described in Section 3.1, and show that higher learning rates lead to higher KTA even under the central flow. Since the only difference between the central flow trajectories at different learning rates is the differing sharpness penalty,[3] this experiment supports our contention that sharpness reduction leads to higher KTA. In Figure 4b, starting at various points along the central flow trajectory for $\eta = 0.1$, we branch off and run gradient flow $\frac{dW}{dt} = -\eta \nabla L(W)$ for 100 units of time (red). We find that gradient flow takes a trajectory with lower KTA than the central flow. Since gradient flow differs from the central flow only by omitting the sharpness penalty term, this experiment further supports

---

[3]The $\eta$ prefactor in eq. (3) is just a time rescaling and does not affect the overall trajectory, only the speed at which the flow traverses that trajectory. It is only via the sharpness penalty that $\eta$ influences the trajectory.

our contention that sharpness reduction leads to higher KTA. Results under more settings can be found in Appendix A.7.

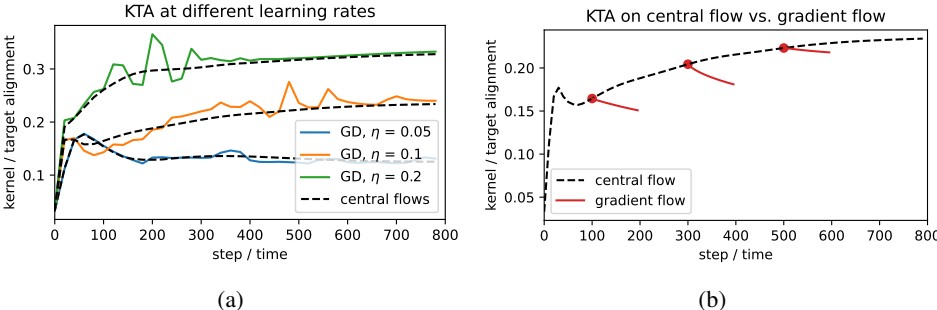

(a)                                        (b)

Figure 4: (a) The KTA under the central flows (dashed black) match those of gradient descent (colors). In particular, the KTA is larger when $\eta$ is larger. (b) Gradient flow (red), when branched off from the central flow at times $\{100, 300, 500\}$, takes a lower-KTA trajectory than the central flow (dashed black). The experiments are conducted in the fully-connected setting described in Section 3.1.

## 6   Conclusion and Future Work

This paper analyzes the evolution of NTK on EoS by examining its alignment effect with the training target. We show that the EoS dynamics enhance the alignment of NTK with the target and provide a theoretical analysis of the underlying mechanism of this phenomenon on a 2-layer linear network. There are several potential future directions: 1. It is known that the evolution of NTK is related to feature learning. Hence it would be interesting to connect our findings to more feature learning properties. 2. Although we provide a detailed theoretical analysis on 2-layer linear networks and partially generalize it to 2-layer ReLU networks in Appendix A.6, a more general theoretical understanding of multi-layer nonlinear networks is still needed in the future.

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

# A Supplementary Empirical Results

## A.1 Experimental details

Here we would like to provide more details of our experimental setup.

**Training resources**  The experiment on the linear network (see Section 3.1 for details) was trained on a CPU since the model is small. All the other experiments, including those described in Section 3.1 and additional experiments presented later in Appendix A.2, were conducted on V100 GPUs with 40 GB of memory.

**Dataset**  The main dataset we use is CIFAR-10 [KH+09]. As is described in Section 3.1, due to GPU memory constraints, we randomly selected about 2000 examples in classes 0 and 1 (roughly 1000 for each class) and relabeled the target values as $\pm 1$. In Appendix A.2.3 we will use a subset of the IMDB dataset [MDP+11] where we randomly selected 1024 examples.

**Network architectures**  Here we provide more details on the network architectures we use.

1. Linear network. As described in Section 3.1, we use a 2-layer linear network with input dimension 400 on Gaussian data. The size of the hidden layer is also 400.

2. Fully connected network. We trained a 4-layer fully connected network with GeLU activation. The input dimension is $32 * 32 * 3$, which is the size after flattening a CIFAR-10 image. The sizes of the three hidden layers are 128,128,64, respectively.

3. Vision Transformer. We also trained a simple Vision Transformer (ViT) using an online implementation[4]. We use their "SimpleViT" model with detailed parameters SimpleViT(image_size=32, patch_size = 4, num_classes = 1, dim = 64, depth = 4, heads = 8, mlp_dim = 256).

4. VGG-11. We also conducted experiments on VGG[Sim14]. We use an online implementation[5] with the "vgg11_bn" architecture in their implementation.

5. ResNet-20. In Appendix A.2.2, we conducted experiments on a ResNet[HZRS16] with 20 layers and GeLU activation. We use Group Normalization instead of Batch Normalization in this ResNet.

6. BERT-small. In Appendix A.2.3 We conducted experiments on language tasks where we fine-tuned BERT-small [TCLT19, BDR21] from an online implementation[6].

## A.2 Experiments in more settings

### A.2.1 VGG

On the same dataset as in Section 3.1, we also trained a VGG-11 network using GD under MSE loss from the same initialization under different learning rates to roughly the same training loss. We again plotted the evolution of KTA (every 20 iterations) in Figure 5a and the individual eigenvector target alignment for the final models in 5b, where we see the increase of KTA and the alignment shift phenomenon. Similar to Section 3.1, we also observe a consistent correlation between a fast increase in KTA and the sharpness-decreasing period of EoS (see Figure 5c) and provide a quantitative analysis in Table 3 to show this connection. Similar to Section 5, we also study the above connection in this VGG setting using the central flow[CDT+24] in Appendix A.7 and demonstrate the benefit of the sharpness penalty to the increase of KTA.

### A.2.2 ResNet

On the same dataset as in Section 3.1, we also trained a ResNet with 20 layers using GeLU activation under GD and MSE loss from the same initialization. Again, we trained the model under different learning rates to roughly the same training loss. We plotted the evolution of KTA (every 20 iterations) in Figure 6a and the individual eigenvector target alignment (along with its cumulative summation) for the final models in Figure 6b, where we see the increase of KTA and the alignment shift. In

---

[4]`https://github.com/lucidrains/vit-pytorch`. MIT license.

[5]`https://github.com/chengyangfu/pytorch-vgg-cifar10`. MIT license.

[6]`https://huggingface.co/docs/transformers/v4.16.2/en/training`. Apache-2.0 license.

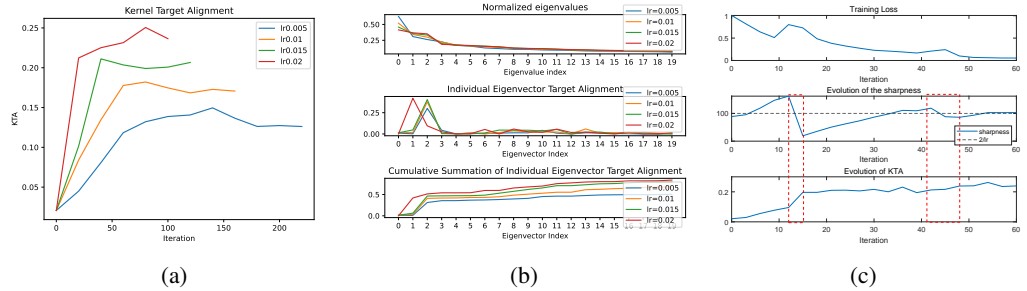

(a)            (b)            (c)

Figure 5: (a) KTA across different learning rates, (b) the alignment shift, and (c) connection between the alignment dynamics and Edge of Stability in VGG-11 (see Section A.2.1). Similar to Figure 1b, we plot the alignment of the first 20 eigenvectors in (b).

Table 3: Comparison of the changing speed of KTA between the sharpness-decreasing periods and sharpness-increasing periods on VGG-11. See Appendix A.2.1 for the detailed experimental setup.

| Iteration interval | 12-15 | 15-42 | 42-48 | 48-78 |
|---|---|---|---|---|
| sharpness behavior | decrease | increase | decrease | gradually stablized |
| total change of KTA | 0.1008 | 0.0163 | 0.0260 | 0.0083 |
| avg change of KTA per step | 0.0336 | 0.0006 | 0.0043 | 0.0003 |

Section A.7, we study the connection between the increase of KTA and EoS using the central flow[CDT⁺24] and demonstrate the benefit of the sharpness penalty to the increase of KTA.

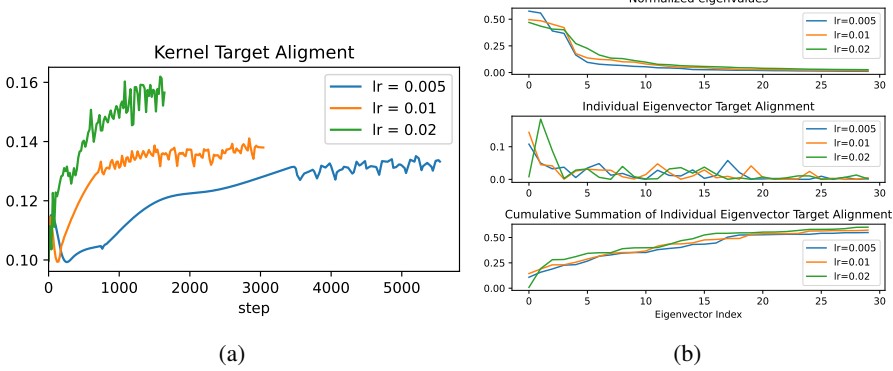

(a)            (b)

Figure 6: (a) KTA across different learning rates, (b) the alignment shift in ResNet-20. More details can be found in Appendix A.2.2.

### A.2.3 Language transformer

In this section, we provide experiments to demonstrate that our results on the alignment behavior also generalize to language tasks. We fine-tuned BERT-small [TCLT19, BDR21] on a subset (1024 randomly selected examples) of the IMDB dataset [MDP⁺11]. The task is to classify whether movie reviews are positive or negative[7], and the labels are ±1. Again, we trained the model from the same initialization under different learning rates to roughly the same training loss. We plotted the evolution of KTA (every 20 steps) in Figure 7a and the cumulative summation of the individual eigenvector target alignment for the final models in Figure 7b. Here, we see the increase of KTA and the alignment shift phenomenon, similar to other tasks.

---

[7]https://huggingface.co/docs/transformers/v4.16.2/en/training

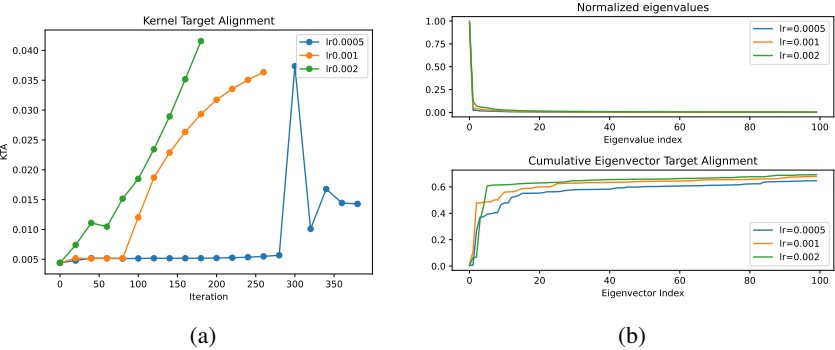

|   |   |
|:-:|:-:|
| (a) | (b) |

Figure 7: (a) KTA across different learning rates, (b) the alignment shift in the sentence classification task using BERT-small. Detailed setup is described in Appendix A.2.3.

### A.2.4 Multi-class tasks

Section 3 presents empirical results on binary classes. Here we provide more experiments on multi-class tasks. More precisely, we randomly selected 2000 examples in all ten classes in CIFAR-10 and trained a VGG-11 network using one-hot labels and MSE loss. We computed the cumulative summation of the individual eigenvector target alignment amd made a comparison after the same *effective* steps: 400 (or 200) steps for a learning rate of 0.01 v.s. 200 (or 100) steps for a learning rate of 0.02, as is shown in Figure 8. Here we see that a larger learning rate has a larger cumulative alignment value, which indicates the alignment shift towards earlier eigenvectors and is consistent with the alignment shift phenomenon in the binary-class case.

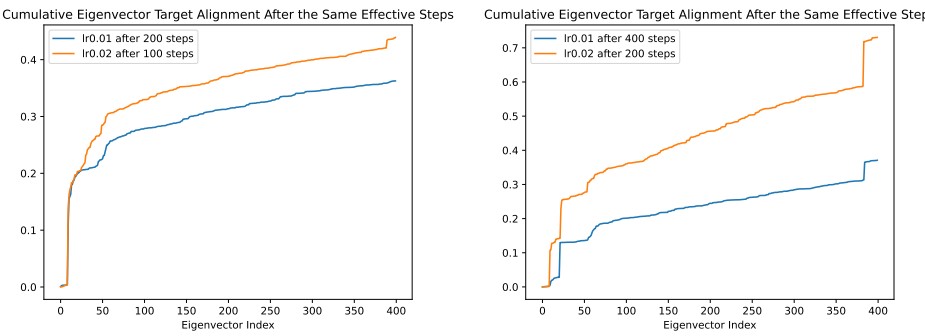

Figure 8: Cumulative summation of the Individual Eigenvector Target Alignment values after the same effective steps on VGG-11 with CIFAR-10.

### A.3 Generalization to the SGD setting

This paper mainly focuses on full-batch GD, since the dynamics of GD are simpler than SGD, and the EoS of GD is well defined. In particular, for GD, the EoS period often consists of distinct phases (e.g., the four phases discussed in our theoretical analysis in Section 4) where the sharpness rises above and then falls below 2/(step size). However, for SGD, [CKL+21] pointed out that "the sharpness does not always settle at any fixed value, let alone one that can be numerically predicted from the hyperparameters". That means for SGD, it's hard to link the increase in KTA to the period where the sharpness falls, as there is no clear division of the cycles and the so-called sharpness-decreasing phase. Nevertheless, we find that our observation that larger learning rates result in stronger KTA alignment still holds for SGD. We added the following experiments on the fully connected setting described in Section 3.1 with batch sizes 1024 and 256. Again, we trained the models under different learning rates to roughly the same training loss. Figure 9a and Figure 10a show the increasing trend of KTA where larger learning rates lead to larger KTA values. Figure 9b and Figure 10b plot the

individual eigenvector target alignment for the final models as well as their cumulative summations, where we see the alignment shift phenomenon.

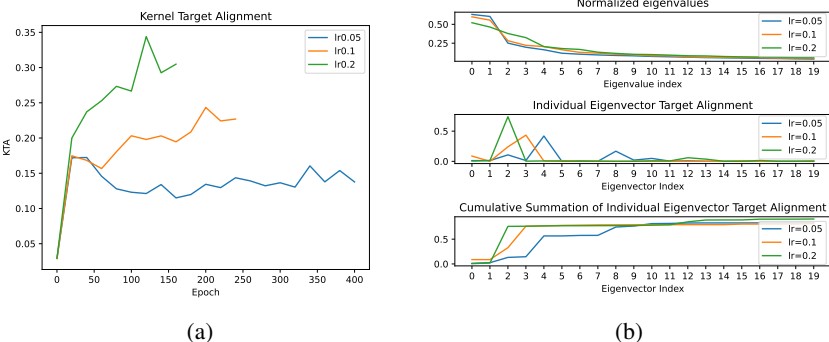

(a)                                                          (b)

Figure 9: SGD experiments in the fully connected network described in Section 3.1 with batch size 1024: (a) KTA across different learning rates, (b) the alignment shift. Similar to Figure 1b, we plot the alignment of the first 20 eigenvectors in (b).

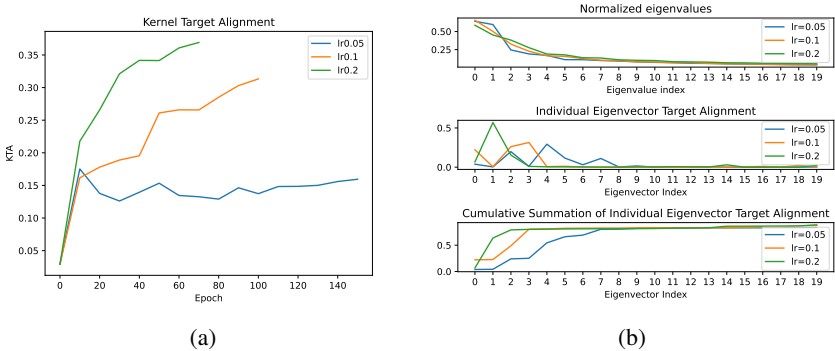

(a)                                                          (b)

Figure 10: SGD experiments in the fully connected network described in Section 3.1 with batch size 256: (a) KTA across different learning rates, (b) the alignment shift. Similar to Figure 1b, we plot the alignment of the first 20 eigenvectors in (b).

### A.4    Comparison with the AGOP alignment in [ZLRB23]

Here we provide evidence on the claim in Section 2 that on the two-layer linear network setting, the enhanced AGOP alignment is less correlated with large learning rates and sharpness reduction in EoS than the enhanced NTK alignment. We trained the 2-layer linear network described in Section 3.1 using GD with learning rates 0.005 and 0.01 and plotted the evolution of the AGOP alignment and KTA conditioned on the same effective iterations (defined as the learning rate times the number of iterations). Figure 11a shows the evolution of these two alignment values during the whole training procedure and Figure 11b zooms in the EoS period where we see the sudden increase of KTA but no significant increase of AGOP.

### A.5    Connection to the generalization ability

As is discussed in Section 3, [ADH+19] proposed a generalization bound proportional to $\sqrt{\frac{y^T K^{-1} y}{n}}$ where $K$ is the NTK and $n$ is the number of the training examples. This can be viewed as an inverse version of the KTA. Here we focus on the 2-layer linear network setting described in Section 3.1 and plot the evolution of the normalized $y^T K^{-1} y$. We demonstrate that the sharpness reduction period of EoS and the alignment shift phenomenon towards earlier eigenvectors are correlated with a sudden decrease of the normalized $y^T K^{-1} y$, see Figure 12a. Figure 12b demonstrates that larger learning rates lead to a more significant decrease of the normalized $y^T K^{-1} y$ than smaller learning rates.

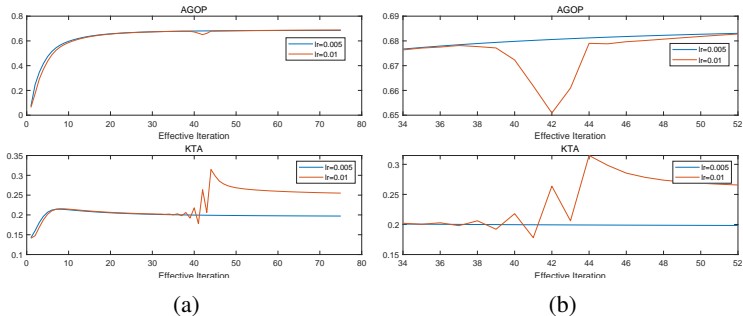

(a)            (b)

Figure 11: Comparison between the AGOP alignment and KTA in a 2-layer linear network described in Section 3.1

We also conducted experiments on the fully-connected network described in Section 3.1. Here, we notice that the tail NTK eigenvalues are much smaller than earlier ones, as is shown in Figure 13a, which can make $K^{-1}$ "noisy" and cause numerical issues. Hence we add $\epsilon I$ to the original $K$ and then calculate the inverse. More formally, we compute the following "inverse KTA": $\frac{y^T (K+\epsilon I)^{-1} y}{\|y\|_2^2 \|(K+\epsilon I)^{-1}\|_F}$. We set $\epsilon = 0.0005\lambda_{\max}(K)$. The top and middle subfigures of Figure 13b plot the training loss and the evolution of the sharpness. The bottom subfigure of Figure 13b shows the change of the inverse KTA every 3 steps, where we see negative values, and that means the inverse KTA is decreasing over time. By comparing the middle and bottom subfigures, we see that the sharpness-decreasing periods correspond to larger decreasing values of the inverse KTA than the sharpness-increasing periods.

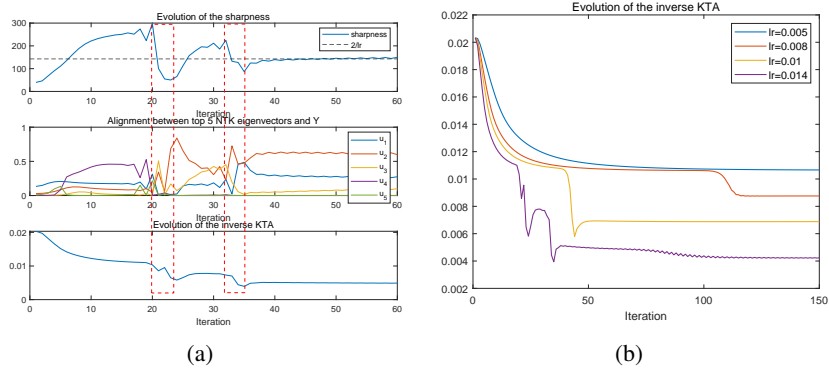

(a)            (b)

Figure 12: (a) Evolution of the sharpness, alignment between the target and the top 5 NTK eigenvectors, and alignment between the target and inverse NTK. (b) The evolution of the alignment between the inverse NTK and the target under different learning rates.

Now we provide empirical results on a direct connection between the alignment shift phenomenon and the generalization ability. Figure 14a and 14b show the alignment shift phenomenon under different learning rates where we see that a larger learning rate leads to a more significant sharpness oscillation and a larger alignment shift towards earlier eigenvectors. To compare the generalization ability, we measure the decrease of the test loss in two ways.

1. Measures the test loss $L_{\text{test}}$ after the same effective iterations, i.e. for different learning rates $\eta_1, ..., \eta_n$, train the network for $t_1, ..., t_n$ iterations such that $\eta_1 t_1 = \eta_2 t_2 = ... = \eta_n t_n$.

2. Measures the change of test loss $\Delta L_{\text{test}}$ during the oscillation period when sharpness decreases and the alignment shift phenomenon occurs. For example in Figure 14a, we compute the change between the 55th and the 75th step, and in Figure 14b compute the change between the 10th and the 20th step.

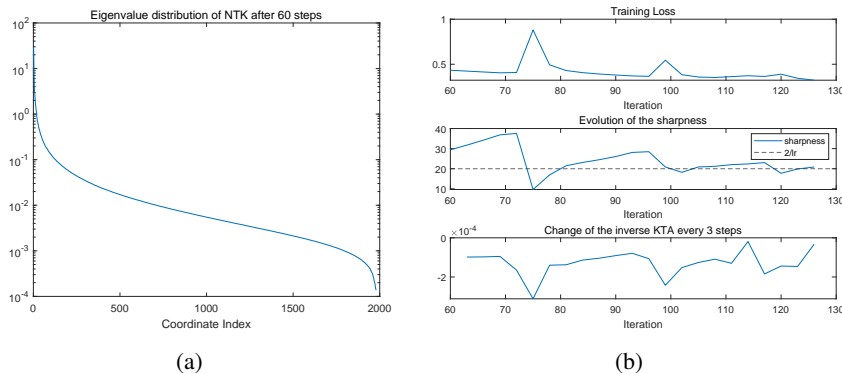

(a)                                        (b)

Figure 13: (a) NTK spectrum in the fully-connected network described in Section 3.1 after 60 steps. The spectra for other steps have roughly the same shape. (b) Evolution of the training loss, the sharpness, and change of the inverse KTA every 3 steps.

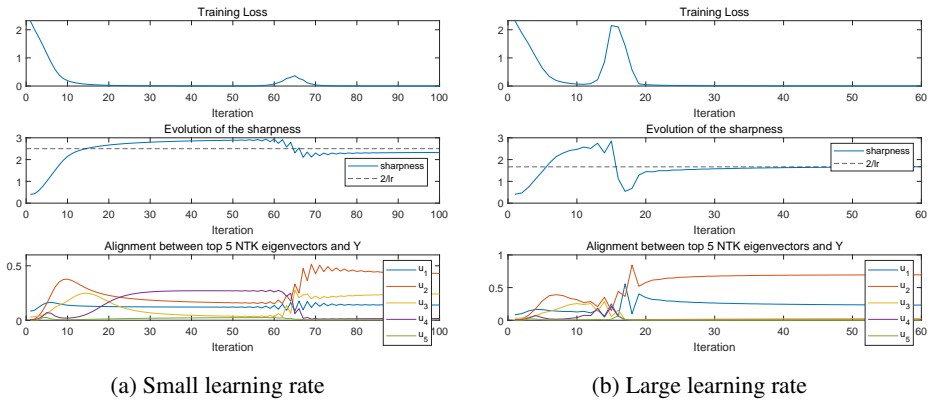

(a) Small learning rate                      (b) Large learning rate

Figure 14: Alignment shift under different learning rates

As is shown in Table 4, a larger learning rate leads to a smaller test loss after the same effective steps, as well as a larger decrease in the test loss during the period when sharpness reduces and the alignment shift phenomenon occurs.

Table 4: Generalization ability under different learning rates

| Learning rate | 0.8 | 0.9 | 1.2 |
|---|---|---|---|
| $L_{\text{test}}$ after the same effective steps | 1.211 | 1.206 | 1.199 |
| $\Delta L_{\text{test}}$ during the oscillation period when alignment shift occurs | -0.008 | -0.020 | -0.062 |

### A.6 Potential Generalizations to Two-layer Nonlinear Networks

In this section, we try to generalize our crucial quantity in Definition 4.6 to the two-layer network with ReLU activation. We again use the MSE loss $L(W) = \frac{1}{2n} \left\| W^{(2)}\sigma(W^{(1)}X) - Y \right\|_F^2$ where $\sigma(\cdot)$ is the elementwise ReLU operator. By simple calculation, the NTK at iteration $t$ in this case is given by

$$K_t = K_t^{(1)} + K_t^{(2)}, \quad \text{where} \quad K_t^{(1)} = \sigma(W_t^{(1)}X)^T\sigma(W_t^{(1)}X), \ K_t^{(2)} = D_t^T D_t,$$

$$\text{with} \quad D_t = \begin{bmatrix} W_t^{(2)}[1]X\text{diag}(1_{\{W_t^{(1)}[1,:]X>0\}}), \\ \cdots, \\ W_t^{(2)}[d_h]X\text{diag}(1_{\{W_t^{(1)}[d_h,:]X>0\}}) \end{bmatrix}.$$

We first demonstrate that $K_t^{(1)} = \sigma(W_t^{(1)}X)^T\sigma(W_t^{(1)}X)$ still exhibits a low-rank structure where it is close to a rank-2 matrix. Here we train a 2-layer ReLU network and plot the eigenvalue distribution of $K_t^{(1)}$ after 5 and 40 steps in Figure 15. We can see that it quickly becomes an approximate rank-2 matrix after only 5 steps.

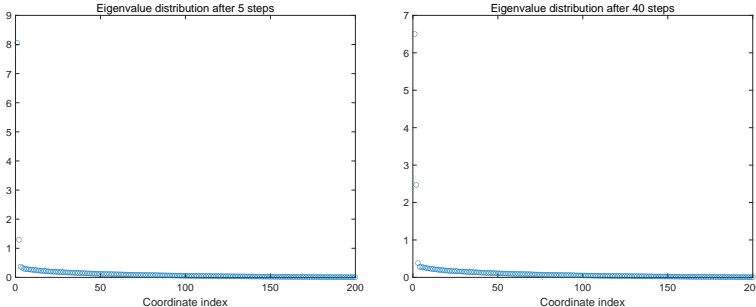

Figure 15: The approximate rank-2 structure of $\sigma(W_t^{(1)}X)^T\sigma(W_t^{(1)}X)$ in a two-layer ReLU network after (a) 5 steps and (b) 40 steps

This approximately rank-2 structure allows us to define a new version of $c_t^2$ as the leading eigenvalue of $K_t^{(2)}$ and $\|\boldsymbol{v}_t\|^2$ as the summation of the top 2 leading eigenvalues of $K_t^{(1)}$, and calculate the corresponding $\alpha_t$ as

$$\alpha_t := \frac{\|\boldsymbol{v}_t\|^2}{c_t^2} = \frac{\lambda_1(K_t^{(1)}) + \lambda_2(K_t^{(1)})}{\lambda_1(K_t^{(2)})}$$

Figure 16 compares the evolution of the sharpness, our new version of $\alpha_t$, and the alignment of $Y$ with top 5 NTK eigenvectors (denoted as $u_1, ..., u_5$) between small and large learning rate settings. The setup is the same as in the linear network setting described in Section 3.1 except that we use ReLU activation now. Here we see that in the large learning rate case, the alignment shift phenomenon happens towards earlier eigenvectors during the sharpness reduction period, and in the meantime, there is a sudden increase of $\alpha_t$. This is consistent with the behavior of $\alpha_t$ and the alignment trend discussed in the theoretical analysis in Section 4.

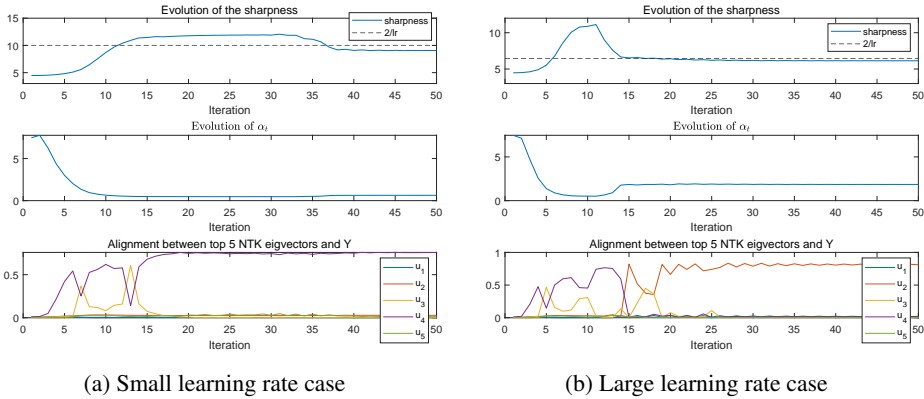

(a) Small learning rate case       (b) Large learning rate case

Figure 16: The alignment shift and evolution of $\alpha_t$ in a two-layer ReLU network

## A.7 Supplementary Results on Central Flow

First, we provide evidence to demonstrate that the central flow is a good approximation to the time-averaged gradient descent. On the fully-connected network, VGG-11 and ResNet-20 described in Section 3.1 and Appendix A.2, we run both central flow and gradient descent under different learning rates. Figure 17 plots the evolution of the training losses, sharpness and the KTA, where we see that central flow approximately matches the time average of GD trajectories. (For the dashed lines in the training loss subplots, we report the "central flow prediction" for the training loss [CDT⁺24], not the

training loss along the central flow.) For the VGG-11 in these experiments, we use average pooling (rather than maxpooling) and GeLU activation (rather than ReLU), so that the training objective is smooth, as is needed in order for the central flows to work well.

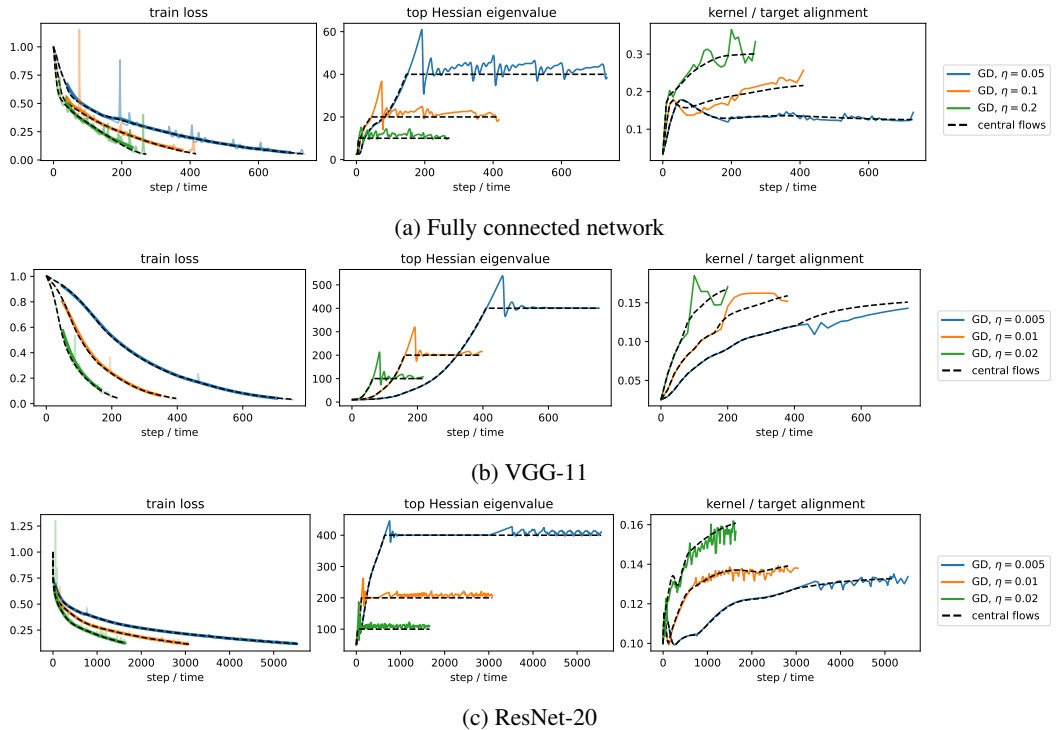

(a) Fully connected network

(b) VGG-11

(c) ResNet-20

Figure 17: Demonstration that the KTA along the central flow reasonably matches that along the GD trajectory under different network architectures. In the subplots on the training losses, the thin/light solid lines are the actual training loss curves under GD, and the thick/dark solid lines are the smoothed (time-averaged) version.

We now provide more results on the evolution of the KTA under central flow compared to gradient flow. Same as in Section 5, in Figure 18. starting at various points along the central flow trajectory in different architectures, we branch off and run gradient flow $\frac{dW}{dt} = -\eta \nabla L(W)$ for some time (red). Again, we find that gradient flow takes a trajectory with lower KTA than the central flow.

# B    Detailed Theoretical Analysis and Proofs for the Four Phases

## B.1    Cartoon Illustration

As is discussed in Section 4.2, the training dynamics at EoS can be divided into four phases based on the position and the sign of change in the sharpness. We plot a cartoon figure to illustrate this division in Figure 19. A similar illustration figure was also provided in [WLL22].

## B.2    Rank-1 structure

In this section, we verify that the structure in Assumption 4.1 will be preserved for $t > t_0$ where $t_0$ is defined in Assumption 4.1.

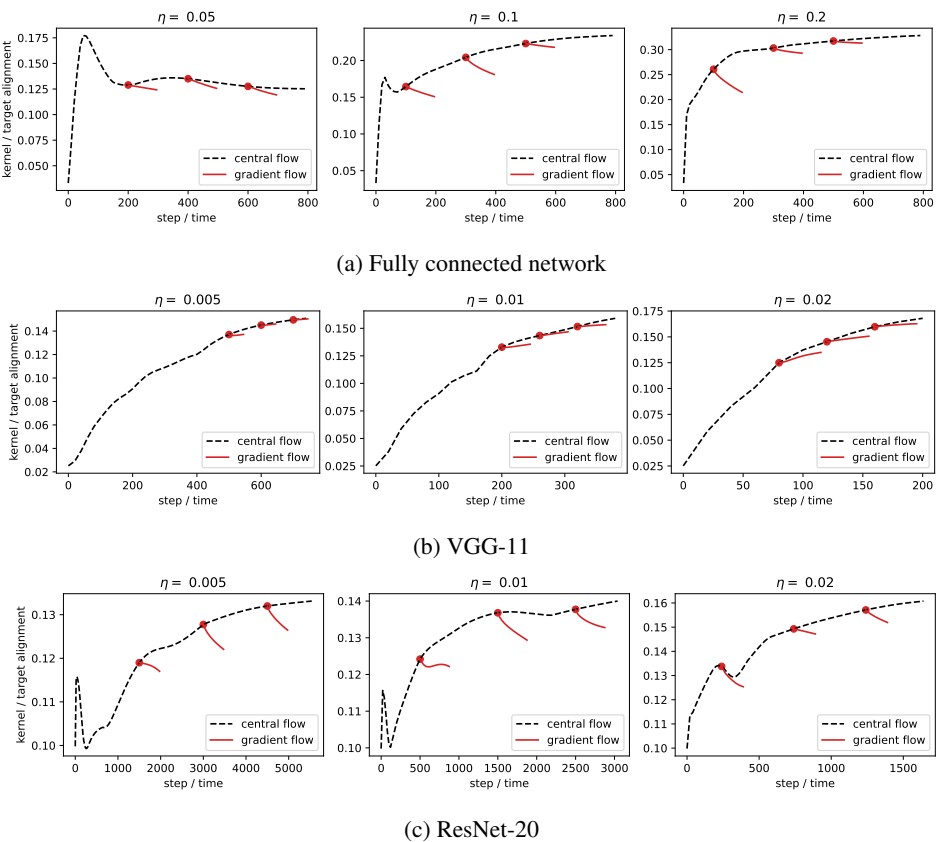

(a) Fully connected network

(b) VGG-11

(c) ResNet-20

Figure 18: Gradient flow (red), when branched off from the central flow at different times, takes a lower-KTA trajectory than the central flow (dashed black).

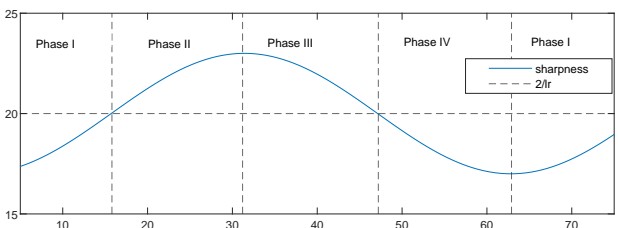

Figure 19: A Cartoon illustration of the four phases during EoS

We prove it by induction. Suppose at time $t$ after $t_0$, we have that $W_t^{(1)}X = \boldsymbol{u}\boldsymbol{v}_t^T$, $W_t^{(2)} = c_t\boldsymbol{u}^T$. Then at time $t+1$, by the update rule of $W_t^{(1)}X$ and $W_t^{(2)}$, we have that:

$$W_{t+1}^{(1)}X = W_t^{(1)}X - \frac{\eta}{n}W_t^{(2)T}E_tX^TX = \boldsymbol{u}\boldsymbol{v}_t^T - \frac{\eta}{n}c_t\boldsymbol{u}E_tXX^T = \boldsymbol{u}(\boldsymbol{v}_t^T - \frac{\eta}{n}c_tE_tXX^T)$$

$$W_{t+1}^{(2)} = W_t^{(2)} - \frac{\eta}{n}E_t(W_t^{(1)}X)^T = c_t\boldsymbol{u}^T - \frac{\eta}{n}E_t\boldsymbol{v}_t\boldsymbol{u}^T = (c_t - \frac{\eta}{n}\langle E_t, \boldsymbol{v}_t\rangle)\boldsymbol{u}^T$$

Hence we see that the rank-1 structure is preserved with $\boldsymbol{u}$ unchanged and

$$c_{t+1} = c_t - \frac{\eta}{n}\langle E_t, \boldsymbol{v}_t\rangle$$

$$\boldsymbol{v}_{t+1}^T = \boldsymbol{v}_t^T - \frac{\eta}{n}c_tE_tXX^T$$

This completes the proof by induction.

## B.3 Analysis of Phase I

Note that under the rank-1 structure in Assumption 4.1, the NTK at iteration $t$ can be written as

$$K_t = \|W_t^{(2)}\|^2 K_x + X^T W_t^{(1)T} W_t^{(1)} X = c_t^2 X^T X + v_t v_t^T$$

We want to prove that during Phase I, both $c_t^2$ and $\|v_t\|^2$ increase over time. We first introduce another assumption on Phase I where we assume that the dynamics follow the GF trajectory. This assumption was also made and empirically verified in [WLL22].

**Assumption B.1.** *During Phase I, the dynamics follow GF trajectory.*

Recall that we denote $F_t = W_t^{(2)} W_t^{(1)} X = c_t v_t$ under the rank-1 structure, and $E_t = F_t - Y$ and use $\{q_i\}_{i=1}^n$ to represent the eigenvectors of $K_x = X^T X$. Then plugging in the update rule of $c_t$ and $v_t$ in Section B.2, we immediately get the following results.

$$\frac{d}{dt} c_t = -\frac{\eta}{n} \langle E_t, v_t \rangle$$

$$\frac{d}{dt} \langle v_t, q_i \rangle = -\frac{\eta}{n} c_t E_t X^T X q_i = -\frac{\eta \lambda_i}{n} c_t \langle E_t, q_i \rangle$$

By Assumption 4.1 we have that at time $t_0$, $c_{t_0} > 0$ and $E_{t_0}[i] = -\Theta(Y[i])$ for all $i \leq n$, which gives us that $\langle E_{t_0}, q_i \rangle = -\Theta(\langle Y, q_i \rangle)$ for all $i \leq n$. That means $\frac{d}{dt} \langle v_t, q_i \rangle$ has the same sign as $\langle Y, q_i \rangle$ at $t = t_0$, which promotes $\langle v_t, q_i \rangle$ to move towards $\langle Y, q_i \rangle$, a direction deviating from $\langle E_t, q_i \rangle$ since $\langle E_t, q_i \rangle$ has the opposite sign as $\langle Y, q_i \rangle$ at $t_0$ and for a short time period later.

If $\langle E_{t_0}, v_{t_0} \rangle < 0$, then $\frac{d}{dt} c_t > 0$ at $t = t_0$, which helps keep the positive sign of $c_t$ for a short time period later. On the other hand, the behavior of $\frac{d}{dt} \langle v_t, q_i \rangle$ will help keep the negative sign of $\langle E_t, v_t \rangle$. Repeating this procedure, we will get a consistent increase of $c_t$ and a consistent movement of $\langle v_t, q_i \rangle$ towards $\langle Y, q_i \rangle$ for a short time period after $t_0$.

If $\langle E_{t_0}, v_{t_0} \rangle > 0$, then $\frac{d}{dt} c_t < 0$ at $t = t_0$, which makes $c_t$ decrease for a short time period later. Note that in this case, the behavior of $\frac{d}{dt} \langle v_t, q_i \rangle$ will promote $\langle E_t, v_t \rangle$ to decrease and become negative during the following training time. Repeating this procedure, we have that:

1) If $\langle E_t, v_t \rangle$ becomes negative before $c_t$ changes the sign, then $c_t$ will start to increase and keep positive and we will go back to the case with $\langle E_t, v_t \rangle < 0$ and $c_t > 0$.

2) If $c_t$ becomes negative before $\langle E_t, v_t \rangle$ changes the sign, then $\langle v_t, q_i \rangle$ will start to move towards $-\langle Y, q_i \rangle$, the same sign as $\langle E_t, q_i \rangle$, which will prevent $\langle E_t, v_t \rangle$ from decreasing and changing the sign. Then $\langle E_t, v_t \rangle$ will keep positive, $c_t < 0$ and will keep decreasing, which will in turn promote $\langle v_t, q_i \rangle$ to move towards $-\langle Y, q_i \rangle$. If we rewrite $-c_t, -v_t$ as $c_t$ and $v_t$ respectively, then we essentially get the same dynamics as in the case with $\langle E_t, v_t \rangle < 0$ and $c_t > 0$.

In summary, the dynamics will eventually move to a stage where we have an increase of $c_t$ and a movement of $\langle v_t, q_i \rangle$ towards $\langle Y, q_i \rangle$, which has the opposite sign as $\langle E_t, q_i \rangle$. Now we assume that at some time point $t' > t_0$ in Phase I, for all $i \leq n$, $\langle v_{t'}, q_i \rangle$ will have the opposite sign as $\langle E_{t'}, q_i \rangle$.

**Assumption B.2.** *Assume that at some time $t'$ in Phase I, $c_{t'} > 0$ and that $\langle v_{t'}, q_i \rangle$ has the opposite sign as $\langle E_{t'}, q_i \rangle$ for all $i \leq n$.*

The following lemma ensures that for later training time $t > t'$ in phase I, before $|\langle E_t, q_i \rangle|$ becomes very small, it always holds that $c_t > 0$ and that $\langle v_t, q_i \rangle$ has the opposite sign as $\langle E_t, q_i \rangle$ for all $i \leq n$. When $|\langle E_t, q_i \rangle|$ becomes very small, it already implies convergence of $E_t$ along $q_i$ direction. The detailed analysis is beyond the scope of this paper and has been studied in [WLL22]. They show that we will either enter EoS or have a small $\|E_t\|$ which implies convergence.

**Lemma B.3.** *Under Assumption B.1,B.2, for $t > t'$ with $t'$ defined in Assumption B.2. For $i \leq n$, when $|\langle E_t, q_i \rangle|$ is not very small, we will have that $c_t > 0$ and that $\langle v_t, q_i \rangle$ has the opposite sign as $\langle E_t, q_i \rangle$. Then we immediately get that $\langle F_t, q_i \rangle = \langle c_t v_t, q_i \rangle$ has the opposite sign as $\langle E_t, q_i \rangle$.*

*Proof.* We have that

$$\frac{d}{dt}\langle E_t, q_i\rangle = -\frac{\eta\lambda_i}{n}c_t^2\langle E_t, q_i\rangle - \frac{\eta}{n}\langle E_t, \boldsymbol{v}_t\rangle\langle \boldsymbol{v}_t, q_i\rangle$$

$$\frac{d}{dt}c_t = -\frac{\eta}{n}\langle E_t, \boldsymbol{v}_t\rangle$$

$$\frac{d}{dt}\langle \boldsymbol{v}_t, q_i\rangle = -\frac{\eta}{n}c_t E_t X^T X q_i = -\frac{\eta\lambda_i}{n}c_t\langle E_t, q_i\rangle$$

Suppose for some $T > t'$ in Phase I such that for all $t \in [t', T]$, $c_t > 0$ and that $\langle E_t, q_i\rangle$ doesn't change the sign for all $i \leq n$. By the definition of $t'$, we know that $\forall i \leq n : \langle \boldsymbol{v}_{t'}, q_i\rangle$ has the opposite sign as $\langle E_{t'}, q_i\rangle$. Then the dynamic of $\langle \boldsymbol{v}_t, q_i\rangle$ gives us that $\langle \boldsymbol{v}_t, q_i\rangle$ has the opposite sign as $\langle E_t, q_i\rangle$ for $t \in [t', T]$. Hence $\langle E_t, \boldsymbol{v}_t\rangle < 0$. Then for $t \in [T, T + dt]$ we have that

$$c_{t+dt} = c_t - \frac{\eta}{n}\langle E_t, \boldsymbol{v}_t\rangle dt > 0$$

and that $\langle E_t, \boldsymbol{v}_t\rangle\langle \boldsymbol{v}_t, q_i\rangle = d_{i,t}\langle E_t, q_i\rangle$ for some $d_{i,t} > 0$. When $|\langle E_t, q_i\rangle|$ is not very small, we will have a bounded $d_{i,t}$, which yields

$$\frac{d}{dt}\langle E_t, q_i\rangle = -\frac{\eta\lambda_i}{n}(c_t^2 + d_{i,t})\langle E_t, q_i\rangle$$

For $t \in [T, T + dt]$ we have that

$$\langle E_{t+dt}, q_i\rangle = \langle E_t, q_i\rangle - \frac{\eta\lambda_i}{n}(c_t^2 + d_{i,t})dt\langle E_t, q_i\rangle = \exp\left(-\frac{\eta\lambda_i}{n}(c_t^2 + d_{i,t})dt\right)\langle E_t, q_i\rangle$$

which has the same sign as $\langle E_t, q_i\rangle$. Processing the above steps completes the proof. $\square$

Now notice that we have the following dynamics for $c_t^2$ and $\|\boldsymbol{v}_t\|^2$,

$$\frac{d}{dt}(c_t^2) = 2c_t\frac{d}{dt}c_t = -\frac{2\eta}{n}\langle E_t, F_t\rangle = -\frac{2\eta}{n}\sum_{i=1}^{n}\langle E_t, q_i\rangle\langle F_t, q_i\rangle$$

$$\frac{d}{dt}\|\boldsymbol{v}_t\|^2 = 2\boldsymbol{v}_t^T\frac{d}{dt}\boldsymbol{v}_t = -\frac{2\eta}{n}\sum_{i=1}^{n}\lambda_i\langle E_t, q_i\rangle\langle F_t, q_i\rangle$$

Then Lemma B.3 implies that $c_t^2$ and $\|\boldsymbol{v}_t\|^2$ are increasing for $t > t'$ in Phase I before $|\langle E_t, q_i\rangle|$ becomes very small. As mentioned above, a more detailed analysis in [WLL22] gives us that we will either enter EoS due to the increase of $c_t^2$ or get convergence before EoS.

## B.4    Intuition on the choice of the learning rate

Let's consider the time period after $c_t^2\lambda_1/n$ reaches $\Theta(1/\eta)$ (denoted as $T_0$). From the proof of Lemma B.3, we have that with $d_{i,t} > 0$,

$$\frac{d}{dt}\langle E_t, q_i\rangle = -\frac{\eta\lambda_i}{n}(c_t^2 + d_{i,t})\langle E_t, q_i\rangle < -\frac{\eta\lambda_i}{n}c_t^2\langle E_t, q_i\rangle = -\Theta(\frac{\lambda_i}{\lambda_1})\langle E_t, q_i\rangle$$

Then after $T = \mathcal{O}(\frac{\lambda_1}{\lambda_i}\log n)$ steps, $\langle E_t, q_i\rangle$ will shrink by $\mathcal{O}(\frac{1}{\text{poly}(n)})$. Let's consider some index $k'$ such that $\lambda_{k'} = \Theta(n^{a+\epsilon})$ for some small $\epsilon > 0$. Note that for $i \leq k'$, $\frac{\lambda_1}{\lambda_i} = \mathcal{O}(n^{1-a-\epsilon})$, hence after $T = \mathcal{O}(n^{1-a-\epsilon}\log n)$ steps, $\langle E_{t+1}, q_i\rangle, i \leq k'$ will shrink by $\mathcal{O}(\frac{1}{\text{poly}(n)})$.

That means after $T = \mathcal{O}(n^{1-a-\epsilon}\log n)$ steps, $\langle -E_t, q_i\rangle\langle F_t, q_i\rangle$ for $i \leq k'$ will be negligible compared to those $i > k'$. Moreover, by our data distribution (see Section 4.1), we know that $k' = \mathcal{O}(\log n)$ and $k = \mathcal{O}(\log n)$ (where $k$ is defined in our data distribution), which gives us $k - k' = \mathcal{O}(\log n)$ and hence the number of terms $\langle -E_t, q_i\rangle\langle F_t, q_i\rangle$ for $k' < i \leq k$ will be negligible compared to the number of terms when $i > k$. Combining the above two facts yields that, $\sum_{i=1}^{n}\lambda_i\langle -E_t, q_i\rangle\langle F_t, q_i\rangle$ will be dominated by $\sum_{i>k}\lambda_i\langle -E_t, q_i\rangle\langle F_t, q_i\rangle$. Therefore,

$$\frac{d}{dt}\|\boldsymbol{v}_t\|^2 = \frac{2\eta}{n}\sum_{i=1}^{n}\lambda_i\langle -E_t, q_i\rangle\langle F_t, q_i\rangle = \frac{2\eta}{n}\Theta(\sum_{i>k}\lambda_i\langle -E_t, q_i\rangle\langle F_t, q_i\rangle)$$

$$= \frac{2\eta}{n}n^a\Theta(\sum_{i>k}\langle -E_t, q_i\rangle\langle F_t, q_i\rangle) = \Theta(\frac{n^a}{n})\frac{d}{dt}(c_t^2\lambda_1)$$

Denote the end of Phase I as $T_1$. Then we have

$$\|\boldsymbol{v}_{T_1}\|^2 - \|\boldsymbol{v}_{T_0+T}\|^2 = \Theta(\frac{n^a}{n})(c_{T_1}^2 - c_{T_0+T}^2)\lambda_1 = \Theta(\frac{n^a}{\eta})$$

Here we check that $T_0 + T$ will not exceed the length of Phase I, $T_1$. Note that

$$\frac{d}{dt}(c_t^2) = -\frac{2\eta}{n}\langle E_t, F_t\rangle \le \frac{\eta}{n}\mathcal{O}(\|Y\|^2) = \mathcal{O}(\eta)$$

Starting from $T_0$ to $T_1$, $c_t^2$ has a total increase of order $\Theta(\frac{1}{\eta})$, which means that $T_1 = T_0 + \Omega(\frac{1}{\eta^2})$. By the choice of learning rate discussed below, we have that $\eta^2 = \Theta(\frac{1}{n^{1-a}})$ and hence $T_1 = T_0 + \Omega(n^{1-a})$, which is at a higher order than $T_0 + T = T_0 + \mathcal{O}(n^{1-a-\epsilon}\log n)$.

**Choice of learning rate** The above calculation implies that the increasing speed of $\|\boldsymbol{v}_t\|^2$ is $\Theta(\frac{n^a}{n})$ times the increasing speed of $c_t^2$ for later period (after $T + T_0$ of Phase I). If we further assume that for early training period, we have $\|\boldsymbol{v}_{T_0+T}\|^2 = \mathcal{O}(\frac{n^a}{\eta})$, then we can get that at the end of Phase I, $\|\boldsymbol{v}_t\|^2 = \Theta(n^a/\eta)$.

Note that at the end of Phase I, i.e. the beginning of EoS, we have $c_t^2 = \Theta(1/\eta)$, then for the output norm, we get that $\|F_t\|^2 = c_t^2\|\boldsymbol{v}_t\|^2 = \Theta(n^a/\eta^2)$. When $\|F_t\|$ goes to order of $\|Y\|$, we want it to go pass the above order $\Theta(n^a/\eta^2)$ so that we will have EoS. (Otherwise $F_t$ will approach $Y$ before reach the order $\Theta(n^a/\eta^2)$ and we will have convergence before EoS) That means we require

$$n^a/\eta^2 \le \mathcal{O}(\|Y\|^2) = \mathcal{O}(n), \quad \Rightarrow \quad \eta^2 = \Omega(\frac{1}{n^{1-a}})$$

So we can pick $\eta^2 = \Theta(\frac{1}{n^{1-a}})$.

## B.5 Dynamics of $c_t^2$

In Phase I under the gradient flow, we prove that the time derivative of $c_t^2$ is determined by the sign of $\langle E_t, F_t\rangle$. Now we prove that in Phase II and later phases, even though gradient descent no longer follows the gradient flow, the sign of $c_{t+1}^2 - c_t^2$ is still determined by the sign of $\langle E_t, F_t\rangle$. Note that the update of $c_t^2$ under gradient descent is given by

$$\begin{aligned}
c_{t+1}^2 - c_t^2 &= -\frac{2\eta}{n}\langle E_t, F_t\rangle + \frac{\eta^2}{n^2}\langle E_t, \boldsymbol{v}_t\rangle^2 = -\frac{2c_t\eta}{n}\langle E_t, \boldsymbol{v}_t\rangle + \frac{\eta^2}{n^2}\langle E_t, \boldsymbol{v}_t\rangle^2 \\
&= \frac{\eta}{n}\langle E_t, \boldsymbol{v}_t\rangle\left(\frac{\eta}{n}\langle E_t, \boldsymbol{v}_t\rangle - 2c_t\right) = \frac{\eta}{n}\langle E_t, \boldsymbol{v}_t\rangle\left(\frac{\eta}{n}\langle c_t\boldsymbol{v}_t - Y, \boldsymbol{v}_t\rangle - 2c_t\right) \\
&= \frac{\eta}{n}\langle E_t, \boldsymbol{v}_t\rangle\left(c_t\left(\frac{\eta}{n}\|\boldsymbol{v}_t\|^2 - 2\right) - \frac{\eta}{n}\langle Y, \boldsymbol{v}_t\rangle\right)
\end{aligned}$$

By Assumption 4.3, we have that $c_t^2\|\boldsymbol{v}_t\|^2 = \|F_t\|^2 \le 2\|E_t\|^2 + 2\|Y\|^2 \le \mathcal{O}(\frac{n}{\eta})$. Combining with the fact that $c_t^2 = \Theta(\frac{1}{\eta})$ during EoS yields that $\|\boldsymbol{v}_t\|^2 \le \mathcal{O}(n)$. Then we have $\frac{\eta}{n}\|\boldsymbol{v}_t\|^2 - 2 = -\Theta(1)$ and that $\left|\frac{\eta}{n}\langle Y, \boldsymbol{v}_t\rangle\right| \le \frac{\eta}{n}\|Y\|\|\boldsymbol{v}_t\| \le \mathcal{O}(\eta)$. Hence $c_t\left(\frac{\eta}{n}\|\boldsymbol{v}_t\|^2 - 2\right) - \frac{\eta}{n}\langle Y, \boldsymbol{v}_t\rangle = -c_t\Theta(1)$, which implies that the update of $c_t^2$ has the same sign as $-\langle E_t, \boldsymbol{v}_t\rangle c_t = -\langle E_t, F_t\rangle$.

## B.6 Analysis of Phase II and Proof of Lemma 4.4

First note that during Phase II, we have that $\langle E_t, F_t\rangle < 0$, which means that $\|E_t\| \le \|Y\|$ because otherwise, we would have that $\langle E_t, F_t\rangle = \|E_t\|^2 + \langle E_t, Y\rangle \ge \|E_t\|^2 - \|E_t\|\|Y\| > 0$. Similarly, we have that $\|F_t\| \le \|Y\|$. Hence we have $|\langle E_t, F_t\rangle| \le \mathcal{O}(\|Y\|^2) = \mathcal{O}(n)$.

We prove Lemma 4.4 by induction using the following two lemmas.

**Lemma B.4.** *Under Assumption 4.1-4.3 during Phase II and pick $\eta = \Theta(n^{-\frac{1-a}{2}})$. Write $\Delta_t := \frac{\lambda_1}{n}c_t^2 - \frac{2}{\eta}$ with $\Delta_t > 0$. Suppose for $s \le t - 1$, we have $|\langle E_s, F_s\rangle| = \mathcal{O}(\delta_2\|Y\|^2)$, we can get that*

$$|\langle E_t, q_1\rangle| \ge (1 + C\eta\Delta_{t-1})|\langle E_{t-1}, q_1\rangle| - \mathcal{O}(\eta^2\delta_2|\langle Y, q_1\rangle|)$$

$$|\langle E_t, q_i\rangle| \le \mathcal{O}\left(\delta_2\eta^2\frac{\lambda_1}{\lambda_i}|\langle Y, q_i\rangle| + |\langle E_{T_1}, q_1\rangle|\right), \quad 2 \le i \le n$$

*where $T_1$ is the end of Phase I.*

*Proof.* During EoS we have that $\frac{\lambda_1}{n}c_t^2 = \Theta(\frac{1}{\eta})$. Combining with $\lambda_1 = \Theta(n)$ assumed in our data distribution (see Section 4.1) gives us that $c_t^2 = \Theta(\frac{1}{\eta})$. For the update of $\langle E_t, q_i \rangle$, we have that

$$
\begin{aligned}
\langle E_t, q_i \rangle &= \langle E_{t-1}, q_i \rangle + \langle E_t - E_{t-1}, q_i \rangle = \langle E_{t-1}, q_i \rangle + \langle c_t \boldsymbol{v}_t - c_{t-1}\boldsymbol{v}_{t-1}, q_i \rangle \\
&= \langle E_{t-1}, q_i \rangle + (c_t - c_{t-1})\langle \boldsymbol{v}_t, q_i \rangle + c_{t-1}\langle \boldsymbol{v}_t - \boldsymbol{v}_{t-1}, q_i \rangle + (c_t - c_{t-1})\langle \boldsymbol{v}_t - \boldsymbol{v}_{t-1}, q_i \rangle \\
&= \langle E_{t-1}, q_i \rangle - \frac{\eta}{n}\langle E_{t-1}, \boldsymbol{v}_{t-1} \rangle \langle \boldsymbol{v}_{t-1}, q_i \rangle - \frac{\eta \lambda_i}{n}c_{t-1}^2\langle E_{t-1}, q_i \rangle \\
&\quad + \frac{\eta^2 \lambda_i}{n^2}\langle E_{t-1}, F_{t-1} \rangle \langle E_{t-1}, q_i \rangle \\
&= \langle E_{t-1}, q_i \rangle - \frac{\eta}{nc_{t-1}^2}\langle E_{t-1}, F_{t-1} \rangle \langle F_{t-1}, q_i \rangle - \frac{\eta \lambda_i}{n}c_{t-1}^2\langle E_{t-1}, q_i \rangle \\
&\quad + \frac{\eta^2 \lambda_i}{n^2}\langle E_{t-1}, F_{t-1} \rangle \langle E_{t-1}, q_i \rangle \\
&= \left( 1 - \frac{\eta \lambda_i}{n}c_{t-1}^2 - \frac{\eta}{nc_{t-1}^2}\langle E_{t-1}, F_{t-1} \rangle + \frac{\eta^2 \lambda_i}{n^2}\langle E_{t-1}, F_{t-1} \rangle \right) \langle E_{t-1}, q_i \rangle \\
&\quad - \frac{\eta}{nc_{t-1}^2}\langle E_{t-1}, F_{t-1} \rangle \langle Y, q_i \rangle
\end{aligned}
\tag{4}
$$

For $i = 1$, we have that

$$
\begin{aligned}
\langle E_t, q_1 \rangle &\overset{(i)}{=} \left( 1 - 2 - \eta \Delta_{t-1} + \frac{\eta^2 \lambda_1(1 + \eta \Delta_{t-1})}{n^2(2 + \eta \Delta_{t-1})}\langle E_{t-1}, F_{t-1} \rangle \right) \langle E_{t-1}, q_1 \rangle \\
&\quad - \frac{\eta}{nc_{t-1}^2}\langle E_{t-1}, F_{t-1} \rangle \langle Y, q_1 \rangle \\
&\overset{(ii)}{=} \left( -1 - \eta \Delta_{t-1} - \frac{\eta^2 \lambda_1(1 + \eta \Delta_{t-1})}{n^2(2 + \eta \Delta_{t-1})}|\langle E_{t-1}, F_{t-1} \rangle| \right) \langle E_{t-1}, q_1 \rangle \\
&\quad - \Theta\left( \frac{\eta^2}{n} \right)\langle E_{t-1}, F_{t-1} \rangle \langle Y, q_1 \rangle
\end{aligned}
$$

where $(i)$ uses $\frac{\lambda_1}{n}c_{t-1}^2 = \frac{2}{\eta} + \Delta_{t-1}$ and $(ii)$ uses $\frac{\lambda_1}{n}c_{t-1}^2 = \Theta(\frac{1}{\eta})$ and $\lambda_1 = \Theta(n)$ and the fact that during phase II, $\langle E_{t-1}, F_{t-1} \rangle < 0$.

Then by the assumption $|\langle E_{t-1}, F_{t-1} \rangle| = \mathcal{O}(\delta_2 \|Y\|^2) = \mathcal{O}(\delta_2 n)$, it immediately follows that

$$
|\langle E_t, q_1 \rangle| \geq (1 + \eta \Delta_{t-1})|\langle E_{t-1}, q_1 \rangle| - \mathcal{O}(\eta^2 \delta_2 |\langle Y, q_1 \rangle|)
$$

Now we bound $|\langle E_s, q_i \rangle|$ for any $s \leq t - 1$ with $i \geq 2$. First notice that $\frac{\eta \lambda_i}{n}c_{s-1}^2 = \Theta(\frac{\lambda_i}{\lambda_1})$ and $\frac{\eta^2 \lambda_i}{n^2}|\langle E_{s-1}, F_{s-1} \rangle| = \mathcal{O}(\frac{\eta^2 \delta_2 \lambda_i}{n^2}\|Y\|^2) = \Theta(\frac{\eta^2 \delta_2 \lambda_i}{\lambda_1})$ by the assumption $|\langle E_{s-1}, F_{s-1} \rangle| \leq \mathcal{O}(\delta_2 \|Y\|^2) = \mathcal{O}(\delta_2 n)$, then we can merge these two terms together and write

$$
-\frac{\eta \lambda_i}{n}c_{s-1}^2 + \frac{\eta^2 \lambda_i}{n^2}\langle E_{s-1}, F_{s-1} \rangle = -\Theta\left( \frac{\lambda_i}{\lambda_1} \right)
\tag{5}
$$

Substituting into eq. (4) and replacing $t$ by $s$ give us

$$
\begin{aligned}
|\langle E_s, q_i \rangle| &\overset{(iii)}{\leq} \left( 1 - \Theta\left( \frac{\lambda_i}{\lambda_1} \right) + \Theta\left( \frac{\eta^2}{n} \right)|\langle E_{s-1}, F_{s-1} \rangle| \right) |\langle E_{s-1}, q_i \rangle| \\
&\quad + \Theta\left( \frac{\eta^2}{n} \right)|\langle E_{s-1}, F_{s-1} \rangle||\langle Y, q_i \rangle| \\
&\overset{(iv)}{\leq} \left( 1 - \Theta\left( \frac{\lambda_i}{\lambda_1} \right) + \mathcal{O}(\eta^2 \delta_2) \right) |\langle E_{s-1}, q_i \rangle| + \Theta(\eta^2 \delta_2)|\langle Y, q_i \rangle| \\
&\overset{(v)}{\leq} \left( 1 - \Theta\left( \frac{\lambda_i}{\lambda_1} \right) \right) |\langle E_{s-1}, q_i \rangle| + \Theta(\eta^2 \delta_2)|\langle Y, q_i \rangle|
\end{aligned}
$$

where $(iii)$ substitutes the order $c_{s-1}^2 = \Theta(\frac{1}{\eta})$, $(iv)$ uses $|\langle E_{s-1}, F_{s-1}\rangle| \leq \mathcal{O}(\delta_2 n)$ again and $(v)$ is because under our data distribution (see Section 4.1) and the choice of learning rate $\eta = \Theta(n^{-\frac{1-a}{2}})$, we have $\eta^2\delta_2 = \Theta(\frac{n^a}{n}\delta_2) \leq \mathcal{O}(\frac{\lambda_i}{\lambda_1}\delta_2) = o(\frac{\lambda_i}{\lambda_1})$. Telescoping this from $T_1$ to $t$ completes the proof.

$\square$

**Lemma B.5.** *Under the conditions of Lemma B.4, we have that at time $t$, $|\langle E_t, F_t\rangle| = \mathcal{O}(\delta_2\|Y\|^2)$*

*Proof.* The upper bound of $|\langle E_t, q_i\rangle|$ for $i \geq 2$ in Lemma B.4 can be further upper bounded by

$$\mathcal{O}\left(\delta_2\eta^2\frac{\lambda_1}{\lambda_i}\sqrt{\frac{\lambda_i}{\sum_k \lambda_k}}\|Y\| + |\langle E_t, q_1\rangle|\right) = \mathcal{O}\left(\delta_2\frac{\eta^2 n}{\sqrt{\lambda_i}}\frac{\|Y\|}{\sqrt{n^{1+a}}} + |\langle E_t, q_1\rangle|\right)$$

which gives us

$$\sum_{i=2}^n \langle E_t, q_i\rangle^2 \leq \sum_{i=2}^n \mathcal{O}\left(\frac{\delta_2^2\eta^4 n^2}{\lambda_i n^{1+a}}\|Y\|^2 + \langle E_{T_1}, q_i\rangle^2\right)$$

$$\leq \mathcal{O}\left(\delta_2^2\eta^4 n^{2-2a}\right)\|Y\|^2 + \|E_{T_1}\|^2 = \mathcal{O}(\delta_2^2\|Y\|^2)$$

where the last equality uses $\eta = \Theta(n^{-\frac{1-a}{2}})$ and that $\|E_{T_1}\|^2 = \delta_0^2\|Y\|^2 = \mathcal{O}(\delta_2^2\|Y\|^2)$.

By the analysis of Phase I, we know that $|\langle E_t.q_i\rangle|$ for $2 \leq i \leq k$ shrink faster than those for $i > k$ in Phase I. Hence at $T_1$, we have that

$$\sum_{i=2}^k \lambda_i\langle E_{T_1}, q_i\rangle^2 \leq \frac{\lambda_1 k}{n-k}\sum_{i>k}\langle E_{T_1}, q_i\rangle^2 = \mathcal{O}(\frac{\lambda_1\log n}{n})\sum_{i>k}\langle E_{T_1}, q_i\rangle^2 = \mathcal{O}(\log n)\sum_{i>k}\langle E_{T_1}, q_i\rangle^2$$

Therefore,

$$\sum_{i=2}^n \lambda_i\langle E_{T_1}, q_i\rangle^2 \leq \sum_{i>k}\mathcal{O}(\log n + \lambda_i)\langle E_{T_1}, q_i\rangle^2 \leq \mathcal{O}(n^a\|E_{T_1}\|^2) = \mathcal{O}(\delta_0^2 n^a\|Y\|^2)$$

Then for time $t$ in Phase II, we can get that

$$\sum_{i=2}^n \lambda_i\langle E_t, q_i\rangle^2 \leq \sum_{i=2}^n \mathcal{O}\left(\frac{\delta_2^2\eta^4 n^2}{n^{1+a}}\|Y\|^2 + \lambda_i\langle E_{T_1}, q_i\rangle^2\right)$$

$$= \mathcal{O}\left(\delta_2^2\eta^4 n^{2-a} + \delta_0^2 n^a\right)\|Y\|^2 = \mathcal{O}(\delta_2^2 n^a)\|Y\|^2$$

(6)

We have that $\delta_1 = \mathcal{O}\left(\sqrt{\frac{\lambda_1}{\sum_k \lambda_k}}\right) = \mathcal{O}(n^{-a/2})$ and that $\sin(Y, q_1) = \sqrt{1-\delta_1^2} = 1 - \Theta(\delta_1^2)$.

In the subspace spanned by $q_1$ and $Y$, choose $\hat{Y} := Y/\|Y\|$ and $\hat{p}$ as orthonormal basis. We have that

$$\|q_1 - \hat{p}\|^2 = 2 - 2\cos(q_1, \hat{p}) = 2 - 2\sin(q_1, Y) = \Theta(\delta_1^2)$$

Hence we know that

$$\langle E_t, \hat{p}\rangle^2 = \langle E_t, q_1\rangle^2 - 2\langle E_t, q_1\rangle\langle E_t, q_1 - \hat{p}\rangle + \langle E_t, q_1 - \hat{p}\rangle^2$$

$$\geq \langle E_t, q_1\rangle^2 - 2|\langle E_t, q_1\rangle|\|E_t\|\|q_1 - \hat{p}\| - \|E_t\|^2\|q_1 - \hat{p}\|^2$$

$$\geq \langle E_t, q_1\rangle^2 - \Theta(\delta_1)\|E_t\||\langle E_t, q_1\rangle| - \Theta(\delta_1^2)\|E_t\|^2$$

$$\geq \langle E_t, q_1\rangle^2 - \Theta(\delta_1)\|E_t\|^2$$

$$\Rightarrow \langle E_t, \hat{Y}\rangle^2 \leq \|E_t\|^2 - \langle E_t, \hat{p}\rangle^2 \leq \|E_t\|^2 - \langle E_t, q_1\rangle^2 + \Theta(\delta_1)\|E_t\|^2$$

$$= \sum_{i=2}^n \langle E_t, q_i\rangle^2 + \Theta(\delta_1)\|E_t\|^2$$

$$\leq \mathcal{O}(\delta_2^2\|Y\|^2) + \mathcal{O}(\delta_1\|Y\|^2) \leq \mathcal{O}(\delta_2^2\|Y\|^2)$$

In the subspace spanned by $E_t$ and $Y$, denote $\hat{Y}$ and $\hat{r}_t$ as orthonormal basis. Note that $F_t = E_t + Y$ also lies in this subspace. Since $\langle E_t, F_t\rangle < 0$, by geometry relationships between $E_t, Y, F_t$ (note

that they form a triangle with $\cos(E_t, F_t) < 0$), we know that $|\langle E_t, \hat{r}_t\rangle\langle F_t, \hat{r}_t\rangle| \leq |\langle E_t, \hat{Y}\rangle\langle F_t, \hat{Y}\rangle|$, which yields that

$$
\begin{aligned}
|\langle E_t, F_t\rangle| &\leq |\langle E_t, \hat{r}_t\rangle\langle F_t, \hat{r}_t\rangle| + |\langle E_t, \hat{Y}\rangle\langle F_t, \hat{Y}\rangle| \\
&\leq 2|\langle E_t, \hat{Y}\rangle\langle F_t, \hat{Y}\rangle| \leq \mathcal{O}(|\langle E_t, \hat{Y}\rangle|\|Y\|) \leq \mathcal{O}(\delta_2\|Y\|^2)
\end{aligned}
$$

$\square$

Now we are ready to prove Lemma 4.4. Combining Lemma B.4 and B.5 together, and noting that at the beginning of phase II, we have $|\langle E_t, F_t\rangle| \leq \|E_t\|\|F_t\| \leq \mathcal{O}(\delta_0\|Y\|^2) \leq \mathcal{O}(\delta_2\|Y\|^2)$ by Assumption 4.3, we can prove by induction that Lemma 4.4 holds for $t$ in Phase II.

By the proof details of Lemma B.5, we know that $|\langle E_t, \hat{Y}\rangle| \leq \mathcal{O}(\delta_2\|Y\|)$. This allows us to prove that $t$ in Phase II,

$$
\cos(Y, \boldsymbol{v}_t) = \cos(Y, F_t) = \frac{\langle F_t, \hat{Y}\rangle}{\|F_t\|} \geq \frac{\langle Y, \hat{Y}\rangle + \langle E_t, \hat{Y}\rangle}{\|Y\|} \geq 1 - \frac{|\langle E_t, \hat{Y}\rangle|}{\|Y\|} \geq 1 - \mathcal{O}(\delta_2)
$$

## B.7 Analysis of Phase III and Proof of Theorem 4.8 Part (A)

Since in Phase III, we have $\langle E_t, F_t\rangle < 0$, then by the dynamics of $c_t^2$ discussed in Appendix B.5, we know that $c_{t+1}^2 < c_t^2$.

Now we analyze the dynamics of $\|\boldsymbol{v}_t\|^2$. We start by introducing the following lemma on the dynamics of $\langle E_t, q_i\rangle$.

**Lemma B.6.** *Under Assumption 4.1-4.3 during Phase III and pick $\eta = \Theta(n^{-\frac{1-a}{2}})$. Write $\Delta_t := \frac{\lambda_1}{n}c_t^2 - \frac{2}{\eta}$ with $\Delta_t > 0$. Then we have that*

$$
|\langle E_t, q_1\rangle| \geq (1 + C_2\eta\Delta_{t-1})|\langle E_{t-1}, q_1\rangle| - \mathcal{O}(\Delta'_{t-1}|\langle Y, q_1\rangle|)
$$

$$
|\langle E_t, q_i\rangle| \leq \left(1 - \Theta(\Delta'_{t-1}) - \Theta\left(\frac{\lambda_i}{\lambda_1}\right)\right)|\langle E_{t-1}, q_i\rangle| + \mathcal{O}(\Delta'_{t-1}|\langle Y, q_i\rangle|), \quad 2 \leq i \leq n
$$

*where $\Delta'_t = \frac{\eta^2}{n}\langle E_t, F_t\rangle$.*

*Proof.* The proof of this lemma is similar to that of Lemma B.4. In particular, eq. (4) still holds. For $i = 1$, we have that

$$
\begin{aligned}
\langle E_t, q_1\rangle &= \left(1 - 2 - \eta\Delta_{t-1} + \frac{\eta^2\lambda_1(1 + \eta\Delta_{t-1})}{n^2(2 + \eta\Delta_{t-1})}\langle E_{t-1}, F_{t-1}\rangle\right)\langle E_{t-1}, q_1\rangle \\
&\quad - \frac{\eta}{nc_{t-1}^2}\langle E_{t-1}, F_{t-1}\rangle\langle Y, q_1\rangle \\
&\stackrel{(i)}{=} (-1 - \eta\Delta_{t-1} + \mathcal{O}(\eta))\langle E_{t-1}, q_1\rangle - \Theta\left(\frac{\eta^2}{n}\right)\langle E_{t-1}, F_{t-1}\rangle\langle Y, q_1\rangle
\end{aligned}
$$

where $(i)$ is because under Assumption 4.3 we have that $|\langle E_{t-1}, F_{t-1}\rangle| \leq \mathcal{O}(\frac{n}{\eta})$ and thus

$$
\frac{\eta^2\lambda_1(1 + \eta\Delta_{t-1})}{n^2(2 + \eta\Delta_{t-1})}|\langle E_{t-1}, F_{t-1}\rangle| \leq \mathcal{O}\left(\frac{\eta\lambda_1}{n}\right) = \mathcal{O}(\eta)
$$

By our choice of $\Delta_{t-1}$, we have $\eta\Delta_{t-1} = \Omega(\text{poly}(n))$. Plugging in this and the definition of $\Delta'_t$ gives us that

$$
\begin{aligned}
|\langle E_t, q_1\rangle| &\geq (1 + \eta\Delta_{t-1} - \mathcal{O}(\eta))|\langle E_{t-1}, q_1\rangle| - \mathcal{O}(\Delta'_{t-1}|\langle Y, q_1\rangle|) \\
&\geq (1 + \eta\Delta_{t-1})|\langle E_{t-1}, q_1\rangle| - \mathcal{O}(\Delta'_{t-1}|\langle Y, q_1\rangle|)
\end{aligned}
$$

For $i \geq 2$, although we do not have the assumption $|\langle E_{t-1}, F_{t-1}\rangle| \leq \mathcal{O}(\delta_2\|Y\|^2) = \mathcal{O}(\delta_2 n)$, by Assumption 4.3 we have that $\frac{\eta^2\lambda_i}{n^2}|\langle E_{t-1}, F_{t-1}\rangle| \leq \mathcal{O}(\frac{\eta\lambda_i}{n}) = \Theta(\frac{\eta\lambda_i}{\lambda_1})$, which is at a lower order

than $\frac{\eta\lambda_i}{n}c_{s-1}^2 = \Theta(\frac{\lambda_i}{\lambda_1})$. That means we can still merge these two terms together and eq. (5) still holds. Then it follows that

$$|\langle E_t, q_i\rangle| \leq \left(1 - \Theta\left(\frac{\lambda_i}{\lambda_1}\right) - \Theta\left(\frac{\eta^2}{n}\right)\langle E_{t-1}, F_{t-1}\rangle\right)|\langle E_{t-1}, q_i\rangle| + \Theta\left(\frac{\eta^2}{n}\right)\langle E_{t-1}, F_{t-1}\rangle|\langle Y, q_i\rangle|$$

$$\leq \left(1 - \Theta\left(\frac{\lambda_i}{\lambda_1}\right) - \Theta\left(\Delta'_{t-1}\right)\right)|\langle E_{t-1}, q_i\rangle| + \Theta\left(\Delta'_{t-1}|\langle Y, q_i\rangle|\right)$$

Note that in the last term of the first inequality, we use $\langle E_{t-1}, F_{t-1}\rangle > 0$ in Phase III. $\qquad\square$

The first inequality in the above lemma tells us that $|\langle E_t, q_1\rangle|$ keeps the increasing trend as in Phase II. When it increases to $\Omega(\delta_2\|Y\|)$, we have the following lemma.

**Lemma B.7.** *During time intervals in Phase II and III when $\|E_t\| = \mathcal{O}(\sqrt{n})$ and $\Delta_t \geq \Omega(\sqrt{\delta_1})$, denote $T := \inf\{t : |\langle E_t, q_1\rangle| > \delta_2\|Y\|\}$ and $\delta_t := |\langle E_t, q_1\rangle|/\|Y\|$, then we have that for $t \geq T$, $|\langle E_t, q_i\rangle| \leq \mathcal{O}(\delta_t\eta^2\frac{\lambda_1}{\lambda_i}|\langle Y, q_i\rangle| + |\langle E_{T_1}, q_i\rangle|)$ where $T_1$ is the end of Phase I.*

*Proof.* We first prove that for $t \geq T$, $|\langle E_t, q_1\rangle|$ increases monotonically, which gives us that $\delta_t \geq \delta_2$. To see this, note that both Lemma B.4 and Lemma B.6 give us $|\langle E_{t+1}, q_1\rangle| \geq (1+C_2\eta\Delta_t)|\langle E_t, q_1\rangle| - \mathcal{O}(|\Delta'_t||\langle Y, q_1\rangle|)$ with $\Delta'_t = \frac{\eta^2}{n}\langle E_t, F_t\rangle$. Suppose at time $t \geq T$, $\delta_t \geq \delta_2$, then we have $|\langle E_t, q_1\rangle| \geq \delta_2\|Y\| \geq \frac{1}{\sqrt{\delta_1}}|\langle Y, q_1\rangle|$ by $|\langle Y, q_1\rangle| = \delta_1\|Y\|$ and $\delta_2 \geq \sqrt{\delta_1}$. Combining with the condition $\Delta_t \geq \Omega(\sqrt{\delta_1})$ gives us that $C_2\eta\Delta_t|\langle E_t, q_1\rangle| \geq \Omega(\eta|\langle Y, q_1\rangle|) \geq \Omega(\Delta'_t|\langle Y, q_1\rangle|)$ by $|\Delta'_t| \leq \mathcal{O}(\frac{\eta^2}{n}\cdot\frac{n}{\eta}) \leq \mathcal{O}(\eta)$. Hence $|\langle E_{t+1}, q_1\rangle| \geq |\langle E_t, q_1\rangle|$ and $\delta_{t+1} \geq \delta_2$.

Now we prove this lemma by induction.

Let's first prove the base case $t = T$. For $t \leq T$, if $T$ is in Phase II, then we know that Lemma B.5 holds. If $T$ is in Phase III, we can modify the proof of Lemma B.5 to get that Lemma B.5 still holds for $t \leq T$. To see this, note that the only part we need to modify is the part using the geometry relationship between the vectors $E_t, F_t$ and $Y$. The geometry relationship $|\langle E_t, \hat{r}_t\rangle\langle F_t, \hat{r}_t\rangle| \leq |\langle E_t, \hat{Y}\rangle\langle F_t, \hat{Y}\rangle|$ doesn't hold in Phase III because now we have $\langle E_t, F_t\rangle > 0$. However, the inequality $\sum_{i=2}^n\langle E_t, q_i\rangle^2 \leq \mathcal{O}(\delta_2^2\|Y\|^2)$ still holds. That means when $t \leq T$, i.e. when $\langle E_t, q_1\rangle^2 \leq \delta_2^2\|Y\|^2$, we have that $\|E_t\|^2 = \sum_{i=2}^n\langle E_t, q_i\rangle^2 \leq \mathcal{O}(\delta_2^2\|Y\|^2)$ and hence $\|E_t\| \leq \mathcal{O}(\delta_2\|Y\|)$. Then we get that $\|F_t\| \leq \|E_t\| + \|Y\| \leq \mathcal{O}(\|Y\|)$ and that $|\langle E_t, F_t\rangle| \leq \|E_t\|\|F_t\| \leq \mathcal{O}(\delta_2\|Y\|^2)$. That means for $t \leq T$, specially $t = T$, the proof of Lemma 4.4 still holds, and we have that $|\langle E_t, q_i\rangle| \leq \mathcal{O}(\delta_2\eta^2\frac{\lambda_1}{\lambda_i}|\langle Y, q_i\rangle| + |\langle E_{T_1}, q_i\rangle|)$. Note that we have proved $\delta_t \geq \delta_2$, then we prove the base case $t = T$.

Now let's deal with the case when $t > T$. Suppose for $T < s \leq t$, we have that $|\langle E_s, q_i\rangle| \leq \mathcal{O}(\delta_s\eta^2\frac{\lambda_1}{\lambda_i}|\langle Y, q_i\rangle| + |\langle E_{T_1}, q_i\rangle|)$. Then applying the calculation in the proof of Lemma B.5, we get that $\sum_{i=2}^n\langle E_s, q_i\rangle^2 \leq \mathcal{O}(\delta_s^2\|Y\|^2)$, which means $\langle E_s, q_1\rangle^2 \geq \Omega(\sum_{i=2}^n\langle E_s, q_i\rangle^2)$. Hence for $T < s \leq t$, $\|E_s\|^2 = \mathcal{O}(\langle E_s, q_1\rangle^2) \leq \mathcal{O}(\langle E_t, q_1\rangle^2) = \mathcal{O}(\delta_t^2\|Y\|^2)$, which implies $\|E_s\| \leq \mathcal{O}(\delta_t\|Y\|)$. Note that $\|E_s\| = \mathcal{O}(\sqrt{n}) = \mathcal{O}(\|Y\|)$, we have that $\|F_s\| \leq \|E_s\| + \|Y\| \leq \mathcal{O}(\|Y\|)$. Therefore, $|\langle E_s, F_s\rangle| \leq \|E_s\|\|F_s\| \leq \mathcal{O}(\delta_t\|Y\|^2)$. Then at time $t + 1$, imitating the calculation in Lemma B.4, we get that $|\langle E_{t+1}, q_i\rangle| \leq \mathcal{O}(\delta_t\eta^2\frac{\lambda_1}{\lambda_i}|\langle Y, q_i\rangle| + |\langle E_{T_1}, q_i\rangle|)$. Note that $\delta_t\eta^2\frac{\lambda_1}{\lambda_i}|\langle Y, q_i\rangle| = \frac{|\langle E_t, q_1\rangle|}{\|Y\|}\eta^2\frac{\lambda_1}{\lambda_i}|\langle Y, q_i\rangle| \leq \mathcal{O}(\frac{|\langle E_{t+1}, q_1\rangle|}{\|Y\|}\eta^2\frac{\lambda_1}{\lambda_i}|\langle Y, q_i\rangle|) = \mathcal{O}(\delta_{t+1}\eta^2\frac{\lambda_1}{\lambda_i}|\langle Y, q_i\rangle|)$. Then we have $|\langle E_{t+1}, q_i\rangle| \leq \mathcal{O}(\delta_t\eta^2\frac{\lambda_1}{\lambda_i}|\langle Y, q_i\rangle| + |\langle E_{T_1}, q_i\rangle|) \leq \mathcal{O}(\delta_{t+1}\eta^2\frac{\lambda_1}{\lambda_i}|\langle Y, q_i\rangle| + |\langle E_{T_1}, q_i\rangle|)$. Hence we can complete the proof by induction. $\qquad\square$

Now we are ready to prove the first part of Theorem 4.8. The goal is to give a sufficient condition of $\Delta_t$ to ensure the increase of $\|v_t\|^2$. Substituting the results in Lemma B.7 to the calculation in Lemma B.5, we know that during time intervals in Phase II and III when $\|E_t\| = \mathcal{O}(\sqrt{n})$ and $\Delta_t \geq \Omega(\sqrt{\delta_1})$, for $t \geq T$, we have $\sum_{i=2}^n\langle E_t, q_i\rangle^2 \leq \mathcal{O}(\delta_t^2\|Y\|^2)$, $\sum_{i=2}^n\lambda_i\langle E_t, q_i\rangle^2 \leq \mathcal{O}(\delta_t^2 n^a)\|Y\|^2$ and that $\sum_{i=2}^n\lambda_i\langle E_t, q_i\rangle\langle Y, q_i\rangle \leq \mathcal{O}(\delta_t\eta^2\lambda_1)\|Y\|^2 = \mathcal{O}(\delta_t n^a)\|Y\|^2$.

Now we start analyzing the update of $\|\boldsymbol{v}_t\|^2$. Recall that we denote $K_x = X^T X$. We have for $\|\boldsymbol{v}_t\|^2$,

$$\|\boldsymbol{v}_{t+1}\|^2 - \|\boldsymbol{v}_t\|^2 = -\frac{2\eta}{n} E_t K_x F_t^T + \frac{\eta^2}{n^2} c_t^2 E_t K_x^2 E_t^T$$

$$= -\frac{2\eta}{n} E_t K_x E_t^T - \frac{2\eta}{n} E_t K_x Y^T + \frac{2\eta + \Delta_t \eta^2}{n} \cdot \frac{E_t K_x^2 E_t^T}{\lambda_1}$$

$$= \frac{2\eta + \Delta_t \eta^2}{n} \cdot \sum_{i=2}^{n} \frac{\lambda_i^2}{\lambda_1} \langle E_t, q_i \rangle^2 + \frac{\Delta_t \eta^2}{n} \lambda_1 \langle E_t, q_1 \rangle^2 - \frac{2\eta}{n} \sum_{i=2}^{n} \lambda_i \langle E_t, q_i \rangle^2 - \frac{2\eta}{n} E_t K_x Y^T$$

We want to have that $\frac{\Delta_t \eta^2}{n} \lambda_1 \langle E_t, q_1 \rangle^2 - \frac{2\eta}{n} \sum_{i=2}^{n} \lambda_i \langle E_t, q_i \rangle^2 - \frac{2\eta}{n} E_t K_x Y^T > 0$, which means that

$$\Delta_t \eta \lambda_1 \langle E_t, q_1 \rangle^2 - 2 \sum_{i=2}^{n} \lambda_i \langle E_t, q_i \rangle^2 - 2\lambda_1 \langle E_t, q_1 \rangle \langle Y, q_1 \rangle - 2 \sum_{i=2}^{n} \lambda_i \langle E_t, q_i \rangle \langle Y, q_i \rangle > 0$$

When $\|E_t\|^2 \leq \mathcal{O}(n)$ and $t \geq T$, we can use Lemma B.7 and get that $\lambda_1 \langle E_t, q_1 \rangle^2 = \Theta(\delta_t^2 n \|Y\|^2) > \lambda_1 \delta_2 |\langle E_t, q_1 \rangle| \|Y\| \geq \frac{1}{\sqrt{\delta_1}} \lambda_1 |\langle E_t, q_1 \rangle \langle Y, q_1 \rangle|$ where we use $|\langle Y, q_1 \rangle| = \delta_1 \|Y\|$ and $\delta_2 \geq \sqrt{\delta_1}$. We also get that $\sum_{i=2}^{n} \lambda_i \langle E_t, q_i \rangle^2 \leq \mathcal{O}(\delta_t^2 n^a) \|Y\|^2$ and $\sum_{i=2}^{n} \lambda_i \langle E_t, q_i \rangle \langle Y, q_i \rangle \leq \mathcal{O}(\delta_t^2 n^a) \|Y\|^2$. Then we need a lower bound of $\Delta_t$: $\Delta_t = \max\{\Omega(\frac{\sqrt{\delta_1}}{\eta}), \Omega(\frac{1}{\eta n^{1-a}})\} = \max\{\Omega(\frac{1}{\eta n^{a/4}}), \Omega(\frac{1}{\eta n^{1-a}})\}$.

When $\|E_t\|^2 = \Omega(n)$, Lemma B.6 tells us that for $i \geq 2$,

$$|\langle E_t, q_i \rangle| \leq \left(1 - \Theta\left(\frac{\lambda_i}{\lambda_1}\right) - \Theta\left(\Delta'_{t-1}\right)\right) \cdot |\langle E_{t-1}, q_i \rangle| + \mathcal{O}(\Delta'_{t-1} |\langle Y, q_i \rangle|)$$

$$\leq \left(1 - \Theta(\Delta'_{t-1})\right) \cdot |\langle E_{t-1}, q_i \rangle| + \mathcal{O}(\Delta'_{t-1} |\langle Y, q_i \rangle|),$$

Denote the start of Phase III, which is also the end of Phase II, as $T_2$. We have proved in Lemma 4.4 that $|\langle E_{T_2}, q_i \rangle| \leq \mathcal{O}(\delta_2 \eta^2 \frac{\lambda_1}{\lambda_i} |\langle Y, q_i \rangle| + |\langle E_{T_1}, q_i \rangle|) = o(|\langle Y, q_i \rangle|)$ by our data distribution (see Section 4.1) and our choice of $\eta$. For $t > T_2$, i.e. in Phase III, $|\langle E_t, q_i \rangle|$ may increase but will not exceed the order $\mathcal{O}(|\langle Y, q_i \rangle|)$. This is because once $|\langle E_t, q_i \rangle| \geq \Omega(|\langle Y, q_i \rangle|)$ such that RHS is smaller than $|\langle E_{t-1}, q_i \rangle|$, then we will have $|\langle E_t, q_i \rangle| < |\langle E_{t-1}, q_i \rangle|$. Hence we will have a stable upper bound $|\langle E_t, q_i \rangle| \leq \mathcal{O}(|\langle Y, q_i \rangle|)$. Hence $\sum_{i=2}^{n} \langle E_t, q_i \rangle^2 \leq \mathcal{O}(\|Y\|^2) = \mathcal{O}(n)$. In other words, when $\|E_t\|^2 = \Omega(n)$, $\langle E_t, q_1 \rangle^2$ dominates $\sum_{i=2}^{n} \langle E_t, q_i \rangle^2$. More precisely, denote $T_3$ as the first time in Phase III when $\|E_t\|^2 > 2 \sum_{i=2}^{n} \langle E_t, q_i \rangle^2$. Consider time $t$ such that $t \geq T_3$, we get that $\langle E_t, q_1 \rangle^2 > \sum_{i=2}^{n} \langle E_t, q_1 \rangle^2$. Then we have $\langle E_t, F_t \rangle = \Theta(\|E_t\|^2) = \Theta(\langle E_t, q_1 \rangle^2)$. Now for $s$ in Phase III such that $s \leq t$, by Lemma B.6, we know that for $i \geq 2$,

$$|\langle E_s, q_i \rangle| \leq \left(1 - \Theta\left(\frac{\lambda_i}{\lambda_1}\right) - \Theta\left(\Delta'_{s-1}\right)\right) \cdot |\langle E_{s-1}, q_i \rangle| + \mathcal{O}(\Delta'_{s-1} |\langle Y, q_i \rangle|)$$

$$\leq \left(1 - \Theta\left(\frac{\lambda_i}{\lambda_1}\right)\right) \cdot |\langle E_{s-1}, q_i \rangle| + \mathcal{O}(\Delta'_{s-1} |\langle Y, q_i \rangle|)$$

Telescoping this inequality from time $T$ defined in Lemma B.7 to $t$ yields that

$$|\langle E_t, q_i \rangle| \leq \mathcal{O}\left(\frac{\max_{T \leq s \leq t} \Delta'_{s-1} \lambda_1}{\lambda_i} |\langle Y, q_i \rangle| + |\langle E_T, q_i \rangle|\right)$$

$$= \mathcal{O}\left(\frac{\max_{T \leq s \leq t} \langle E_s, q_1 \rangle^2 \eta^2 \lambda_1}{n \lambda_i} |\langle Y, q_i \rangle| + |\langle E_T, q_i \rangle|\right)$$

$$\leq \mathcal{O}\left(\frac{\eta^2 \langle E_t, q_1 \rangle^2 \lambda_1}{n \lambda_i} |\langle Y, q_i \rangle| + |\langle E_T, q_i \rangle|\right) = \mathcal{O}\left(\frac{\eta^2 \langle E_t, q_1 \rangle^2}{\lambda_i} |\langle Y, q_i \rangle| + |\langle E_T, q_i \rangle|\right)$$

Analyses before already give us that $\sum_{i=2}^{n} \lambda_i \langle E_T, q_i \rangle^2 \leq \mathcal{O}(\delta_2^2 n^a) \|Y\|^2$ and that $\sum_{i=2}^{n} \lambda_i \langle E_T, q_i \rangle \langle Y, q_i \rangle \leq \mathcal{O}(\delta_2^2 n^a) \|Y\|^2$. Then we have that

$$\sum_{i=2}^{n} \lambda_i \langle E_t, q_i \rangle \langle Y, q_i \rangle \leq \mathcal{O}(\delta_2^2 n^a) \|Y\|^2 + \sum_{i=2}^{n} \mathcal{O}\left(\eta^2 \langle E_t, q_1 \rangle^2 \langle Y, q_i \rangle^2\right)$$

$$\leq \mathcal{O}(\delta_2^2 n^a) \|Y\|^2 + \mathcal{O}\left(\eta^2 \|Y\|^2\right) \langle E_t, q_1 \rangle^2 = \mathcal{O}(\eta^2 n) \langle E_t, q_1 \rangle^2$$

where the last equality uses $\eta^2 = \Theta(n^{-(1-a)})$ and $\langle E_t, q_1 \rangle^2 = \Omega(n)$. We also have

$$\sum_{i=2}^{n} \lambda_i \langle E_t, q_i \rangle^2 \leq \mathcal{O}(\delta_2^2 n^a)\|Y\|^2 + \sum_{i=2}^{n} \mathcal{O}\left(\lambda_i \frac{\eta^4 \langle E_t, q_1 \rangle^4}{\lambda_i^2} \langle Y, q_i \rangle^2\right)$$

$$= \mathcal{O}(\delta_2^2 n^a)\|Y\|^2 + \eta^4 \langle E_t, q_1 \rangle^4 \sum_{i=2}^{n} \mathcal{O}\left(\frac{1}{\lambda_i} \langle Y, q_i \rangle^2\right)$$

$$= \mathcal{O}(\delta_2^2 n^a)\|Y\|^2 + \eta^4 \langle E_t, q_1 \rangle^4 \sum_{i=2}^{n} \mathcal{O}\left(\frac{\|Y\|^2}{\sum_k \lambda_k}\right)$$

$$\leq \mathcal{O}(\delta_2^2 n^a)\|Y\|^2 + \eta^4 n^2 \langle E_t, q_1 \rangle^2 \mathcal{O}\left(\frac{\langle E_t, q_1 \rangle^2}{n^{1+a}}\right)$$

$$\overset{(i)}{\leq} \mathcal{O}(\delta_2^2 n^a)\|Y\|^2 + \mathcal{O}(\eta^3 n^{2-a}) \langle E_t, q_1 \rangle^2 \overset{(ii)}{\leq} \mathcal{O}(\eta n) \langle E_t, q_1 \rangle^2$$

where (i) uses Assumption 4.3 that $\langle E_t, q_1 \rangle^2 \leq \|E_t\|^2 \leq \mathcal{O}(n/\eta)$ and (ii) plugs in $\eta^2 = \Theta(n^{-(1-a)})$ and $\langle E_t, q_1 \rangle^2 = \Omega(n)$.

Combining these two cases together yields that

$$\sum_{i=2}^{n} \lambda_i \langle E_t, q_i \rangle^2 + \sum_{i=2}^{n} \lambda_i \langle E_t, q_i \rangle \langle Y, q_i \rangle \leq \mathcal{O}(\max\{\eta, 1\}) \eta \lambda_1 \langle E_t, q_1 \rangle^2 = \mathcal{O}(1) \eta \lambda_1 \langle E_t, q_1 \rangle^2$$

On the other hand, $\lambda_1 \langle E_t, q_1 \rangle \langle Y, q_1 \rangle = \delta_1 \lambda_1 \langle E_t, q_1 \rangle \|Y\| \leq \delta_1 \lambda_1 \langle E_t, q_1 \rangle^2$. That means we require that $\Delta_t \geq \Omega\left(\max\left\{1, \frac{\delta_1}{\eta}\right\}\right) = \max\{\Omega(1), \Omega(\frac{1}{\eta n^{a/2}})\}$.

Combining all the bounds of $\Delta_t$ together and noticing the choice $\eta = \Theta(n^{-\frac{1-a}{2}})$, we can get that $\Delta_t \geq \max\{\Omega(1), \Omega(\frac{1}{\eta n^{a/4}})\}$.

## B.8 Analysis of Phase IV and Proof of Theorem 4.8 Part (B)

By definition of $t_1$ and $t_2$, we know that they are the turning points for $\langle E_t, F_t \rangle$ to change the sign, which means at these two time points, we have $\langle E_{t_1}, F_{t_1} \rangle \approx 0$ and $\langle E_{t_2}, F_{t_2} \rangle \approx 0$. More precisely, by the analysis in Appendix B.6 and Lemma B.5, we know that $\|E_{t_1}\| \leq o(\|Y\|)$ and $|\langle E_{t_1}, F_{t_1} \rangle| \leq \mathcal{O}(\delta_2 \|Y\|^2) = \mathcal{O}(\delta_2)\|F_{t_1}\|^2$. The final part of Appendix B.6 also implies that $\cos(Y, \boldsymbol{v}_{t_1}) \geq 1 - \mathcal{O}(\delta_2)$. Under Assumption 4.7, we know that at time $t_2$, $\|E_{t_2}\| \leq \|E_{t_1}\| = o(\|Y\|)$ and hence $|\langle E_{t_2}, F_{t_2} \rangle| \leq \mathcal{O}(\delta_2 \|Y\|^2) = \mathcal{O}(\delta_2)\|F_{t_2}\|^2 = \mathcal{O}(\delta_2)\|F_{t_1}\|^2$. By a similar argument in the final part of Appendix B.6, we also get that $\cos(Y, \boldsymbol{v}_{t_2}) \geq 1 - \mathcal{O}(\delta_2)$.

Note that $\|Y\|^2 = \|F_t - E_t\|^2 = \|F_t\|^2 + \|E_t\|^2 - 2\langle E_t, F_t \rangle$, we have

$$\|F_{t_2}\|^2 = \|Y\|^2 - \|E_{t_2}\|^2 + 2\langle E_{t_2}, F_{t_2} \rangle \geq \|Y\|^2 - \|E_{t_1}\|^2 + 2\langle E_{t_2}, F_{t_2} \rangle$$

$$= \|F_{t_1}\|^2 + 2\langle E_{t_2}, F_{t_2} \rangle - 2\langle E_{t_1}, F_{t_1} \rangle$$

$$\geq \|F_{t_1}\|^2 - \mathcal{O}(\delta_2)\|F_{t_1}\|^2 = (1 - \mathcal{O}(\delta_2))\|F_{t_1}\|^2$$

Hence if $\Delta_{t_1} \geq C\frac{\delta_2}{\eta}$ for some constant $C > 0$, we can get that

$$\frac{\lambda_1}{n} c_{t_1}^2 - \frac{\lambda_1}{n} c_{t_2}^2 > \frac{\lambda_1}{n} c_{t_1}^2 - \frac{2}{\eta} = \Delta_{t_1} \geq \Omega(\delta_2)\frac{\lambda_1}{n} c_{t_1}^2$$

and hence

$$c_{t_2}^2 = c_{t_1}^2 + c_{t_2}^2 - c_{t_1}^2 \leq (1 - \Omega(\delta_2))c_{t_1}^2$$

which gives us

$$\|\boldsymbol{v}_{t_2}\|^2 = \frac{\|F_{t_2}\|^2}{c_{t_2}^2} \geq \frac{1 - \mathcal{O}(\delta_2)}{1 - \Omega(\delta_2)} \frac{\|F_{t_1}\|^2}{c_{t_1}^2} = \frac{1 - \mathcal{O}(\delta_2)}{1 - \Omega(\delta_2)}\|\boldsymbol{v}_{t_1}\|^2$$

If the constant $C$ in the inequality $\Delta_{t_1} \geq C\frac{\delta_2}{\eta}$ is sufficiently large, we can ensure that $\|\boldsymbol{v}_{t_2}\|^2 > \|\boldsymbol{v}_{t_1}\|^2$ and therefore $\alpha_{t_2} > \alpha_{t_1}$.

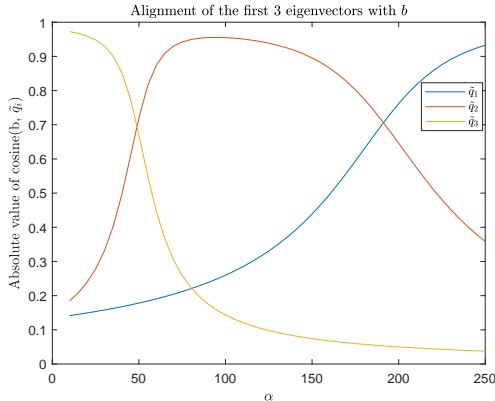

Figure 20: Illustration of Lemma 4.9 in a warm-up setting described in Appendix C.1

## C   Proof of Lemma 4.9

Let $X^T X := Q\text{diag}(\lambda_1, ..., \lambda_n)Q^T$ be the eigenvalue decomposition of the data kernel $X^T X$. The approximate NTK matrix in Lemma 4.9 can be written as $c^2(X^T X + \alpha \hat{\boldsymbol{y}}\hat{\boldsymbol{y}}^T) = c^2 Q(\text{diag}(\lambda_1, ..., \lambda_n) + \alpha(Q^T\hat{\boldsymbol{y}})(Q^T\hat{\boldsymbol{y}})^T)Q^T$. Then it suffices to consider the matrix $A := \text{diag}(\lambda_1, ..., \lambda_n) + \alpha \boldsymbol{b}\boldsymbol{b}^T$ with $\alpha > 0$ and $\boldsymbol{b} := Q^T\hat{\boldsymbol{y}}$. Denote $\tilde{\lambda}_1 \geq \tilde{\lambda}_2 \geq ... \geq \tilde{\lambda}_n$ as the eigenvalues of $A$ with corresponding eigenvectors $\tilde{q}_i$. Let $b[i]$ be the $i$-th component of $\boldsymbol{b}$.

### C.1   Warm-up: A Simple Case

Let's first start with a warm-up case where $\lambda_1 > \lambda_2 > \lambda_3 = \lambda_4 = ... = \lambda_n$, i.e there are only 3 distinct eigenvalues for the data kernel. We first conduct an experiment with $n = 100$, $\lambda_1 = 300, \lambda_2 = 150$ and $\lambda_3 = \lambda_4 = ... = \lambda_n = 100$ and $\boldsymbol{b}$ is a random vector with unit norm. Figure 20 plots the absolute values of the cosine alignment between $\boldsymbol{b}$ and the first 3 leading eigenvectors of $A$, denoted as $\tilde{q}_1, \tilde{q}_2, \tilde{q}_3$. We observe that as $\alpha$ increases, $\boldsymbol{b}$ gradually gets larger alignment values with earlier eigenvectors. We also notice sharp transitions around values 50 and 200, which are equal to $\lambda_2 - \lambda_3$ and $\lambda_1 - \lambda_3$, respectively. Now we start the theoretical analysis in this warm-up setting.

**Assumption C.1.** *Let $A := diag(\lambda_1, \lambda_2, \lambda_3, ..., \lambda_3) + \alpha \boldsymbol{b}\boldsymbol{b}^T \in \mathbb{R}^{n \times n}$ with $\alpha > 0, \|\boldsymbol{b}\|^2 = 1$. We assume that $\lambda_1 - \lambda_2 = \Theta(n), \lambda_2 - \lambda_3 = \Theta(n), \lambda_3 = \Theta(n)$ and $|b[i]| = \Theta(1/\sqrt{n}), \forall i \in [n]$.*

**Lemma C.2.** *Under Assumption C.1, we have that*

1. *When $\alpha < \frac{\lambda_2 - \lambda_3}{1 + \mathcal{O}(n^{-1/2})}$, we have that $\arg\max_{1 \leq i \leq n} |\langle \tilde{q}_i, b \rangle| = 3$.*

2. *When $\frac{\lambda_2 - \lambda_3}{1 - \mathcal{O}(n^{-1/2})} < \alpha < \frac{\lambda_1 - \lambda_3}{1 + \mathcal{O}(n^{-1/2})}$, we have that $\arg\max_{1 \leq i \leq n} |\langle \tilde{q}_i, b \rangle| = 2$.*

3. *When $\alpha > \frac{\lambda_1 - \lambda_3}{1 - \mathcal{O}(n^{-1/2})}$, we have that $\arg\max_{1 \leq i \leq n} |\langle \tilde{q}_i, b \rangle| = 1$.*

*Proof.* By the Bunch–Nielsen–Sorensen formula [BNS78], the eigenvalues of $A$ have the following structures.

1. The first 3 largest eigenvalues $\tilde{\lambda}_1, \tilde{\lambda}_2, \tilde{\lambda}_3$ are the roots of the following secular equation

$$\sum_{k=1}^{n} \frac{b[k]^2}{\lambda_k - t} = \frac{b[1]^2}{\lambda_1 - t} + \frac{b[2]^2}{\lambda_2 - t} + \frac{R^2}{\lambda_3 - t} = -\frac{1}{\alpha}, \quad \text{where } R^2 = 1 - b[1]^2 - b[2]^2 \qquad (7)$$

They also showed that $\lambda_3 < \tilde{\lambda}_3 < \lambda_2 < \tilde{\lambda}_2 < \lambda_1 < \tilde{\lambda}_1$. Note that under Assumption C.1, $R^2 = 1 - \mathcal{O}(\frac{1}{n})$.

2. The other $n - 3$ eigenvalues are all equal to $\lambda_3$.

By Wikipedia[8], the $k$-th component of $\tilde{q}_i$ ($i = 1, 2, 3$) is given by

$$\tilde{q}_i[k] = \frac{b[k]}{(\lambda_k - \tilde{\lambda}_i)N_i}$$

where $N_i$ is a number that makes the vector $\tilde{q}_i$ normalized. Hence it follows that

$$N_i = \sqrt{\sum_{k=1}^{n} \frac{b[k]^2}{(\lambda_k - \tilde{\lambda}_i)^2}} = \sqrt{\frac{b[1]^2}{(\lambda_1 - \tilde{\lambda}_i)^2} + \frac{b[2]^2}{(\lambda_2 - \tilde{\lambda}_i)^2} + \frac{R^2}{(\lambda_3 - \tilde{\lambda}_i)^2}}$$

where $R^2 = 1 - b[1]^2 - b[2]^2$ is defined in eq. (7). Then for the alignment $\langle \tilde{q}_i, b \rangle$ with $i = 1, 2, 3$, We have that

$$\langle \tilde{q}_i, b \rangle = \frac{\sum_{k=1}^{n} \frac{b[k]^2}{\lambda_k - \tilde{\lambda}_i}}{\sqrt{\frac{b[1]^2}{(\lambda_1 - \tilde{\lambda}_i)^2} + \frac{b[2]^2}{(\lambda_2 - \tilde{\lambda}_i)^2} + \frac{R^2}{(\lambda_3 - \tilde{\lambda}_i)^2}}} = \frac{-1/\alpha}{\sqrt{\frac{b[1]^2}{(\lambda_1 - \tilde{\lambda}_i)^2} + \frac{b[2]^2}{(\lambda_2 - \tilde{\lambda}_i)^2} + \frac{R^2}{(\lambda_3 - \tilde{\lambda}_i)^2}}} \tag{8}$$

**Case 1: small $\alpha$.** Since when $\alpha \to 0$, we have $\tilde{\lambda}_i \to \lambda_i$, then we can choose $\alpha$ sufficiently small such that $\tilde{\lambda}_3(1 + \frac{1}{\sqrt{n}}) < \lambda_2$.

Then by Assumption C.1, we know that

$$\frac{b[1]^2}{\lambda_1 - \tilde{\lambda}_3} + \frac{b[2]^2}{\lambda_2 - \tilde{\lambda}_3} = \mathcal{O}(\frac{1}{n^2}) + \mathcal{O}(\frac{1}{n\sqrt{n}}) = \mathcal{O}(\frac{1}{n\sqrt{n}})$$

Hence eq. (7) is approximated by

$$\frac{R^2}{\lambda_3 - \tilde{\lambda}_3} = -\frac{1}{\alpha} + \mathcal{O}(\frac{1}{n\sqrt{n}})$$

Note that $|\frac{R^2}{\lambda_3 - \tilde{\lambda}_3}| \geq |\frac{R^2}{\lambda_3 - \lambda_2}| = \Theta(\frac{1}{n})$, then we can further write the approximated equation as

$$\frac{R^2}{\lambda_3 - \tilde{\lambda}_3}(1 \pm \mathcal{O}(n^{-1/2})) = -\frac{1}{\alpha}$$

which gives us $\tilde{\lambda}_3 = \lambda_3 + \alpha R^2 (1 \pm \mathcal{O}(n^{-1/2})) = \lambda_3 + \alpha(1 \pm \mathcal{O}(n^{-1/2}))$. Then the condition $\tilde{\lambda}_3(1 + n^{-1/2}) < \lambda_2$ can be satisfied for certain choice of $\alpha$ such that $\alpha < \frac{\lambda_2 - \lambda_3}{1 + \mathcal{O}(n^{-1/2})}$. Then by eq. (8), we have

$$|\langle \tilde{q}_3, b \rangle| = \frac{1/\alpha}{\sqrt{\frac{b[1]^2}{(\lambda_1 - \tilde{\lambda}_3)^2} + \frac{b[2]^2}{(\lambda_2 - \tilde{\lambda}_3)^2} + \frac{R^2}{(\lambda_3 - \tilde{\lambda}_3)^2}}} \approx \frac{\frac{R^2}{|\lambda_3 - \tilde{\lambda}_3|}}{\sqrt{\frac{R^2}{(\lambda_3 - \tilde{\lambda}_3)^2}}} = R = 1 - \mathcal{O}(\frac{1}{n})$$

That means $\arg\max_{1 \leq i \leq n} |\langle \tilde{q}_i, b \rangle| = 3$.

**Case 2: medium $\alpha$.** For the choice of $\alpha$ such that $\lambda_2 < \tilde{\lambda}_2(1 - \mathcal{O}(n^{-1/2}))$ and $\tilde{\lambda}_2(1 + \mathcal{O}(n^{-1/2})) < \lambda_1$. Then we know that $\frac{b[1]^2}{\lambda_1 - \tilde{\lambda}_2} + \left|\frac{b[2]^2}{\lambda_2 - \tilde{\lambda}_2}\right| = \mathcal{O}(\frac{1}{n\sqrt{n}})$ while $|\frac{R^2}{\lambda_3 - \tilde{\lambda}_2}| = \Theta(\frac{1}{n})$. Hence eq. (7) is approximated by

$$\frac{R^2}{\lambda_3 - \tilde{\lambda}_2}(1 \pm \mathcal{O}(n^{-1/2})) = -\frac{1}{\alpha}$$

which gives us $\tilde{\lambda}_2 = \lambda_3 + \alpha R^2(1 \pm \mathcal{O}(n^{-1/2})) = \lambda_3 + \alpha(1 \pm \mathcal{O}(n^{-1/2}))$. Then the condition $\lambda_2 < \tilde{\lambda}_2(1 - \mathcal{O}(n^{-1/2}))$ and $\tilde{\lambda}_2(1 + \mathcal{O}(n^{-1/2})) < \lambda_1$ can be satisfied for certain choice of $\alpha$ such that $\frac{\lambda_2 - \lambda_3}{1 - \mathcal{O}(n^{-1/2})} < \alpha < \frac{\lambda_1 - \lambda_3}{1 + \mathcal{O}(n^{-1/2})}$. Again by eq. (8), we have $|\langle \tilde{q}_2, b \rangle| \approx R = 1 - \mathcal{O}(n^{-1})$. Hence $\arg\max_{1 \leq i \leq n} |\langle \tilde{q}_i, b \rangle| = 2$.

**Case 3: large $\alpha$.** For the choice of $\alpha$ such that $\tilde{\lambda}_1 > \lambda_1(1 + \mathcal{O}(n^{-1/2}))$. By a similar argument as in the previous two cases, we know that $\arg\max_{1 \leq i \leq n} |\langle \tilde{q}_i, b \rangle| = 1$. In this case, the condition $\tilde{\lambda}_1 > \lambda_1(1 + \mathcal{O}(n^{-1/2}))$ can be satisfied for certain choice of $\alpha$ such that $\alpha > \frac{\lambda_1 - \lambda_3}{1 - \mathcal{O}(n^{-1/2})}$.

$\square$

---

[8] https://en.wikipedia.org/wiki/Bunch-Nielsen-Sorensen_formula

## C.2 Analysis of General Cases and Proof of Lemma 4.9

Recall that we denote $p_i$ to be the left singular vectors of $X$. Further denote $\sigma_i$ as the $i$-th singular value of $X$. Note that we write $A := \text{diag}(\lambda_1, ..., \lambda_n) + \alpha \boldsymbol{bb}^T$ with $\boldsymbol{b} := Q^T \hat{\boldsymbol{y}}$. By our data distribution (see Section 4.1), we have $b[i] \propto \langle Y, q_i \rangle = \sigma_i \langle \beta, p_i \rangle$, then we know that $\frac{|b[i]|}{|b[j]|} = \Theta(\frac{\sigma_i}{\sigma_j}), \forall i, j \in [n]$, i.e. $\frac{b[i]^2}{b[i]^2} = \Theta(\frac{\lambda_i}{\lambda_j})$. Combining with the order of $\{\lambda_i\}_{i=1}^n$ in our data distribution yields that $b[i]^2 = \mathcal{O}(\frac{n^a}{n^{1+a}}) = \mathcal{O}(\frac{1}{n})$ for $i \geq k$ where the index $k$ is defined in our data distribution. We know that $k = \mathcal{O}(\log_\gamma n)$.

Now we are ready to prove Lemma 4.9. Recall aht we enote $\tilde{\lambda}_1 \geq \tilde{\lambda}_2 \geq ... \geq \tilde{\lambda}_n$ as the eigenvalues of $A$ with corresponding eigenvectors $\tilde{q}_i$. Again, by the Bunch–Nielsen–Sorensen formula [BNS78], $\tilde{\lambda}_j$ ($1 \leq j \leq n$) satisfies the secular equation

$$\sum_{i=1}^n \frac{b[i]^2}{\lambda_i - \tilde{\lambda}_j} = -\frac{1}{\alpha},$$

[BNS78] also proved that $\lambda_j < \tilde{\lambda}_j < \lambda_{j-1}$ for $j > 1$ and $\tilde{\lambda}_1 > \lambda_1$.

By our data distribution (see Section 4.1), for the eigenvalue $\tilde{\lambda}_j$ with $j \leq k - 1$, we have $\tilde{\lambda}_j - \lambda_i = \Theta(\tilde{\lambda}_j), i > k$. Note that we already proved $b[i]^2 = \mathcal{O}(\frac{1}{n})$ for $i \geq k$. Using these two facts, we can replace $\frac{b[i]^2}{\lambda_i - \tilde{\lambda}_j}, i > k$ by $\frac{b[i]^2}{\lambda_k - \tilde{\lambda}_j}$ with approximation error

$$b[i]^2 \left| \frac{1}{\lambda_k - \tilde{\lambda}_j} - \frac{1}{\lambda_i - \tilde{\lambda}_j} \right| = b[i]^2 \cdot \frac{\lambda_k - \lambda_i}{(\tilde{\lambda}_j - \lambda_k)(\tilde{\lambda}_j - \lambda_i)} = \mathcal{O}\left( \frac{\lambda_k - \lambda_i}{\tilde{\lambda}_j^2} \right)$$

Hence we can write the secular equation $\sum_{i=1}^n \frac{b[i]^2}{\lambda_i - \tilde{\lambda}_j} = -\frac{1}{\alpha}$ as

$$\sum_{i=1}^{k-1} \frac{b[i]^2}{\lambda_i - \tilde{\lambda}_j} + \frac{R^2}{\lambda_k - \tilde{\lambda}_j} + \Delta = -\frac{1}{\alpha},$$

where $R^2 = \sum_{i=k}^n b[i]^2$ and that

$$\Delta \leq \sum_{i=k+1}^n b[i]^2 \left| \frac{1}{\lambda_k - \tilde{\lambda}_j} - \frac{1}{\lambda_i - \tilde{\lambda}_j} \right| \leq \mathcal{O}\left( \frac{1}{\tilde{\lambda}_j^2} \right) \left( \lambda_k - \frac{1}{n-k} \sum_{i=k+1}^n \lambda_i \right) = \mathcal{O}\left( \frac{\delta_\lambda n^a}{\tilde{\lambda}_j^2} \right)$$

where the last equality uses the assumption that $|\frac{1}{n-k} \sum_{i=k+1}^n \lambda_i - \lambda_k| = \delta_\lambda n^a$ with $\delta_\lambda = o(1)$.

By the order of $\{\lambda_i\}_{i=1}^n$ in our data distribution, we have that

$$\frac{\sum_{i=1}^{k-1} b[i]^2}{R^2} = \Theta\left( \frac{\sum_{i=1}^{k-1} \lambda_i}{\sum_{i=k}^n \lambda_i} \right) = \mathcal{O}\left( \frac{n \log n}{n^{a+1}} \right) = \tilde{\mathcal{O}}\left( \frac{1}{n^a} \right),$$

and hence

$$\sum_{i=1}^{k-1} b[i]^2 = \tilde{\mathcal{O}}(\frac{1}{n^a}), R^2 = 1 - \tilde{\mathcal{O}}(\frac{1}{n^a}).$$

The order of $\{\lambda_i\}_{i=1}^n$ in our data distribution also tells us that $\forall j \leq k - 1, \lambda_{j-1} - \lambda_j = \Omega(n^a)$. Then for certain choice of $\alpha$ such that $\lambda_j < \tilde{\lambda}_j(1 - n^{-a/2})$ and $\tilde{\lambda}_j(1 + n^{-a/2}) < \lambda_{j-1}$ for certain $j \leq k - 1$, then we have $|\lambda_i - \tilde{\lambda}_j| = \Omega(\tilde{\lambda}_j n^{-a/2})$ for $i \leq k - 1$. Using these facts, we get that

$$\sum_{i=1}^{k-1} \left| \frac{b[i]^2}{\lambda_i - \tilde{\lambda}_j} \right| \leq \sum_{i \leq k-1} b[i]^2 \mathcal{O}\left( \frac{n^{a/2}}{\tilde{\lambda}_j} \right) = \tilde{\mathcal{O}}\left( \frac{1}{\tilde{\lambda}_j n^{a/2}} \right)$$

Hence the approximated secular equation can further be written as

$$\frac{R^2}{\lambda_k - \tilde{\lambda}_j} = -\frac{1}{\alpha} + \mathcal{O}\left( \frac{\delta_\lambda n^a}{\tilde{\lambda}_j^2} \right) + \tilde{\mathcal{O}}\left( \frac{1}{\tilde{\lambda}_j n^{a/2}} \right), \forall j \leq k - 1$$

Recall that by discussion above, we have $\tilde{\lambda}_j - \lambda_k = \Theta(\tilde{\lambda}_j)$ and that $\tilde{\lambda}_j = \Omega(n^a)$. That means $|\frac{R^2}{\lambda_k - \tilde{\lambda}_j}| = \Theta(\frac{1}{\tilde{\lambda}_j})$ and $\frac{\delta_\lambda n^a}{\tilde{\lambda}_j^2} \leq \frac{\delta_\lambda}{\tilde{\lambda}_j}$. This allows us to write the above approximated equation as

$$\frac{R^2}{\lambda_k - \tilde{\lambda}_j}(1 \pm \mathcal{O}(\delta_\lambda + n^{-a/2})) = -\frac{1}{\alpha}, \forall j \leq k - 1$$

We get that $\tilde{\lambda}_j = \lambda_k + \alpha(1 \pm \mathcal{O}(\delta_\lambda + n^{-a/2}))$. Then the condition $\lambda_j < \tilde{\lambda}_j(1 - n^{-a/2})$ and $\tilde{\lambda}_j(1 + n^{-a/2}) < \lambda_{j-1}$ can be satisfied for certain choice of $\alpha$ such that $\frac{\lambda_j - \lambda_k}{1 - \mathcal{O}(\delta_\lambda + n^{-a/2})} < \alpha < \frac{\lambda_{j-1} - \lambda_k}{1 + \mathcal{O}(\delta_\lambda + n^{-a/2})}$.

Moreover,

$$|\langle \tilde{q}_j, v \rangle| = \frac{1/\alpha}{\sqrt{\sum_{i=1}^n \frac{b[i]^2}{(\lambda_i - \tilde{\lambda}_j)^2}}} \approx \frac{\frac{R^2}{|\lambda_k - \tilde{\lambda}_j|}}{\sqrt{\frac{R^2}{(\lambda_k - \tilde{\lambda}_j)^2}}} = R.$$

That means $\arg\max_{1 \leq i \leq n} |\langle \tilde{q}_i, b \rangle| = j$.

