# OpenReview forum: "Understanding the Evolution of the Neural Tangent Kernel at the Edge of Stability"
_NeurIPS.cc/2025/Conference — NeurIPS 2025 poster_

### Official Review · Reviewer_fnJU · 2025-07-01

**Clarity:** 2
**Significance:** 2
**Originality:** 2
**Rating:** 4
**Confidence:** 4

**Summary:**

The paper investigates the behavior of the NTK during the training of neural networks, particularly focusing on the phenomenon known as the EoS. The authors observe that during EoS, the leading eigenvectors of the NTK align more with the training target when larger learning rates are used. They provide empirical evidence across various architectures and theoretically analyze this alignment shift phenomenon for a two-layer linear network.

**Questions:**

See weakness.

**Ethical Concerns:**

["NO or VERY MINOR ethics concerns only"]

**Final Justification:**

I have read the response and found some points are considerable, so I raise my rating.

**Quality:**

2

**Strengths And Weaknesses:**

Strengths

1. The paper presents comprehensive empirical results across different architectures (e.g., linear networks, fully connected networks, Vision Transformers) and datasets, demonstrating the alignment shift phenomenon.

2. The authors provide a detailed theoretical analysis for a two-layer linear network, which is a significant contribution. This analysis offers insights into why the alignment shift occurs during the sharpness reduction phases of EoS, enhancing the understanding of the underlying mechanisms.

Weakness:

1. Some of the theoretical assumptions may be unrealistic, and the authors also do not verify them in the real-world data and network settings, making their application doubtful.

2. The paper primarily focuses on full-batch gradient descent. While the authors mention that the observations generalize to SGD, a more detailed theoretical analysis in the SGD setting would be beneficial, as SGD is more commonly used in practice.

3. Some presentation should be improved. For e.g., the legend and description axes in the Figures are too small.

---

> ### Author Rebuttal · Authors · 2025-07-30
>
> We thank the reviewer for their comments and would like to address their concerns below.
>
> > Some of the theoretical assumptions may be unrealistic, and the authors also do not verify them in the real-world data and network settings, making their application doubtful.
>
> We would like to elaborate more on the intuitions behind our assumptions and argue that although many, they are either mild assumptions or have been made or verified by prior works.
>
> - Data distribution assumptions: In lines 173-178, we explained the intuition of our data distribution. Here, we add more discussions. For the input $X$, we assume that the first several eigenvalues of the data kernel have large gaps while the remaining ones are close to each other. This captures many datasets in practice where its data kernel eigenvalues decrease quickly for the first several and then slowly for later ones. [WLL22] has a similar assumption that "the gap between two adjacent eigenvalues is not very large, and there is a gap between the largest and the second largest eigenvalue." For the target $Y$, we assume that our target is a linear function of the input, which is common in prior theoretical works.
> - Assumption 4.1 assumes that after a short initial period, $W^{(1)}$ converges to a rank-1 matrix whose left singular vector aligns with the right singular vector of $W^{(2)}$.  We explain the intuition behind this assumption in lines 185-198.  We also want to emphasize that we only make this assumption at a short initial time point, which could be far before entering EoS. If this holds at any time $t_0$, it provably holds for all $t>t_0$ (Appendix B.2).
> - Assumption 4.2 is also a common assumption made in prior works. Please see lines 233-235 for more discussions.
> - Assumption 4.3: The first part of it assumes that the loss during training will not grow larger than poly(n) of the initial loss. Here, n is the number of training examples, which is typically large in practice. So, this is a weak assumption. This can also be seen in our loss curves (e.g. Figure 2(c), 5(c)) where the loss during training is not too large compared to the initial loss. The second part of this assumption stipulates that the loss has undergone a nontrivial decrease when it enters EoS. This has been observed and verified by prior EoS papers and captures the fact that the loss usually decreases quickly during the initial training period.
> - Assumption 4.7 is an additional assumption only required in the second point of our theorem. It captures the non-monotonically decreasing behavior of the loss, where the loss after the spike is typically around or smaller than the value before the spike. This phenomenon was also observed and studied by [CKL+ 21] and [WLL22].
>
> [CKL+ 21] Jeremy M. Cohen, Simran Kaur, Yuanzhi Li, J. Zico Kolter, and Ameet Talwalkar. Gradient descent on neural networks typically occurs at the edge of stability. In 9th International Conference on Learning Representations, ICLR 2021,
>
> [WLL22] Zixuan Wang, Zhouzi Li, and Jian Li. Analyzing sharpness along GD trajectory: Progressive sharpening and edge of stability. Advances in Neural Information Processing Systems, 35:9983–9994, 2022.
>
> > While the authors mention that the observations generalize to SGD, a more detailed theoretical analysis in the SGD setting would be beneficial, as SGD is more commonly used in practice.
>
> We agree that in principle, it would be helpful to analyze SGD.  However, the edge of stability dynamics of SGD remains poorly understood -- see [1] for some recent findings.  For example, for SGD, there is no clean four-stage delineation of the EOS dynamics, as there is for GD. For SGD, even theoretically analyzing the eigenvalue evolution is challenging, let alone the evolution of NTK eigenvectors and the alignment with the target. We study GD because we believe that understanding GD is a necessary stepping stone to understanding SGD.
>
> [1] Arseniy Andreyev and Pierfrancesco Beneventano. "Edge of Stochastic Stability: Revisiting the Edge of Stability for SGD."
>
> > Some presentation should be improved. For e.g., the legend and description axes in the Figures are too small.
>
> We thank the reviewer for this helpful suggestion and will increase the corresponding font sizes in the later version.

---

> > ### Comment · Reviewer_fnJU · 2025-08-05
> >
> > Thank you for the reply. I will raise my score.

---

### Official Review · Reviewer_GAQw · 2025-07-02

**Clarity:** 2
**Significance:** 3
**Originality:** 3
**Rating:** 4
**Confidence:** 3

**Summary:**

This paper studies the evolution of the Neural Tangent Kernel (NTK) at the Edge of Stability (EoS) during gradient descent training. It focuses on how the NTK eigenvectors align with the target and how this alignment evolves over time, especially during periods of sharpness reduction. The authors present empirical results across various architectures, showing that larger learning rates lead to greater NTK–target alignment. The paper also provides a theoretical analysis in the case of two-layer linear networks, describing the 4 phases of the sharpness dynamics and highlighting the key quantities that determine the increase in the alignment.

**Questions:**

See weaknesses above, in particular, regarding the assumption, and what happens to the alignment close to optimal learning rates.

**Ethical Concerns:**

["NO or VERY MINOR ethics concerns only"]

**Final Justification:**

The issue on the learning rate is completely resolved. The one about the rank of the weight matrix only partially. Hence, I raised my score from 3 to 4.

**Limitations:**

yes

**Paper Formatting Concerns:**

No concerns

**Quality:**

2

**Strengths And Weaknesses:**

## Strengths

- Relevance: the paper tackles an important problem: understanding the evolution of the NTK at the Edge of Stability (EoS). This is related to the evolution of the NTK, and thus has obvious connections  to the mechanisms of feature learning through alignment with the target.
- The empirical study spans multiple architectures, and provides convincing evidence that larger learning rates correlate with increased kernel-target alignment (KTA).
- In my view, the main theoretical contribution is the description of an interesting theoretical mechanism: that the $\eta^2$ term drives the growth of $\|v_t\|^2$ during the sharpness reduction phases, contributing to alignment shift.

## Weaknesses

- Experiment on the alignment between eigenvectors and the target (Figure 1, 2, 3) should be repeated with a few more learning rates, to see if the alignemnt purely happens in co-occurrences with the model achieving better performances. Hence, the statement that larger learning rate bring more alignment would have to refined/circumstantiated a bit more. What happens for the optimal learning rate?
- The theoretical section (Section 4) is dense and hard to follow. The narrative around the four phases could be streamlined to focus more directly on the connection between sharpness reduction and alignment increase.  I would suggest re-structure it focusing on the connection with the first empirical part: prove that during the sharpness reduction period, the alignment increases, and discuss the role of the learning rate. I think the full description of the 4 phases of the early training dynamics is not necessary to convey the main points of the paper.
- There is a confusing inconsistency in the learning rate scaling: the main text introduces $\eta = \Theta(n^{-(1-a)/2})$, but some lemmas (e.g., Lemma B.4, B.6) refer to $\eta = \Theta(n^{-(1-a)/n})$. This appears to be a typo but adds to the confusion.
- At times it is hard to comment on how realistic the assumptions are, and how necessary they are for the argument to hold. For instance, the preservation of a rank-1 structure of $W^{(1)}X$ for unwhitened data. Unfortunately, there is no experimental or theoretical evidence that the rank 1 assumption holds. It would also be interesting to know whether the assumption of not using whitened data affects the rank.
Also, in the theoretical part the authors transition from gradient flow to discrete GD, introducing $\eta^2$ terms that appear later without clear transition.
- Minor: figures suffer from small font sizes and thin lines, which may hinder readability.
- Minor: the individual eigenvector target alignment plots are on a linear scale, making it hard to assess contributions from low-alignment modes. A log-scale plot would provide a clearer picture of the alignment shift across all modes.
- Minor: the sentence *Note that the alignment values sum up to 1, so they can be regarded as a kind of “distribution”* is a bit informal a not necessary in the definition.

---

> ### Author Rebuttal · Authors · 2025-07-30
>
> We thank the reviewer for their comments and would like to address their concerns below.
>
> > Experiment on the alignment between eigenvectors and the target (Figure 1, 2, 3) should be repeated with a few more learning rates, to see if the alignment purely happens in co-occurrences with the model achieving better performances. Hence, the statement that larger learning rate bring more alignment would have to refined/circumstantiated a bit more. What happens for the optimal learning rate?
>
> We would like to add experiments with more learning rates here. In the fully-connected setting, we tried SGD with batch size 256 under more learning rates. Here we focus on the test performances of the models. We trained the models with different learning rates to the same training loss of 0.01 and calculated the corresponding test losses and KTA.
>
> | Learning rates | 0.02   | 0.025  | 0.03   | 0.035  |
> | -------------- | ------ | ------ | ------ | ------ |
> | test loss      | 0.4290 | 0.4244 | 0.4241 | 0.4664 |
> | KTA            | 0.3607 | 0.3669 | 0.3878 | 0.3432 |
>
> Here we see that lr=0.03 is roughly the optimal learning rate in terms of the test performance, and we do see a correlation between better test performance and better KTA. We would like to argue that this doesn't conflict with our main message, because the goal of this work is to understand the benefit of large learning rates on feature learning and generalization ability. Here we observe that 1) an overlarge learning rate may lead to bad test performance, 2) in the meantime, the alignment gets worse, too. That means our focus on the alignment is a reasonable choice to understand the effect of the learning rate on the generalization ability.
>
> > The theoretical section (Section 4) is dense and hard to follow.
>
> Thank you for this suggestion.  We agree that the organization you suggest may make the paper more streamlined. We would also like to clarify that the full description of the 4 phases is not just in the early training dynamics, but a division of the whole Edge of Stability cycle. We describe the 4 phases in detail, mainly to help readers who are not familiar with Edge of Stability to better understand its dynamics. After introducing the 4 phases, we will be able to clearly define the sharpness reduction period and prove the increase of alignment during this period. We agree that we should highlight the main message of the theoretical analysis in a better way and will consider re-structuring it.
>
> > There is a confusing inconsistency in the learning rate scaling.
>
> This was indeed a typo. It should be $\eta=\Theta(n^{-\frac{1-a}{2}})$ We apologize --- thank you for catching it.
>
> >At times it is hard to comment on how realistic the assumptions are, and how necessary they are for the argument to hold.
>
> For more explanations on the intuition of our assumptions, please see the first point of the response to Reviewer fnJU.
>
> > For instance, the preservation of a rank-1 structure of $W^{(1)}X$ for unwhitened data. Unfortunately, there is no experimental or theoretical evidence that the rank 1 assumption holds. It would also be interesting to know whether the assumption of not using whitened data affects the rank.
>
> In lines 185-198 of the submission, we give a theoretical argument for why the rank-1 structure should hold, where lines 194-198 discuss the case with unwhitened data. The key reasons here are that 1) under small initialization, the network output is small during the early training period, making the error vector close to $-Y$, 2) we consider the one-dimensional output case where $W^{(2)}$ is a row vector (rank 1). Whether we use whitened or unwhitened data doesn't matter. We also want to emphasize that we only make this rank 1 assumption at a short initial time point, far before entering EoS. If this holds at any time $t_0$, it provably holds for all $t>t_0$ (Appendix B.2).
>
> Now we add experimental evidence on the approximate rank 1 structure. In the 2-layer linear network considered in the theoretical analysis, we train the model under lr=0.01 and calculate the eigenvalues of $W^{(1)T}W^{(1)}$. The top 6 eigenvalues at initialization are [3.9297, 3.8144, 3.8057, 3.7141, 3.6817, 3.6167], where there is no low-rank structure. The top 6 eigenvalues after 20 steps (early training) and 60 steps (later training) are [13.0442, 3.9242, 3.8099, 3.7926, 3.7138, 3.6643] and [20.9646, 3.9208, 3.8095, 3.7950, 3.7115, 3.6542] respectively, where we see the trend to become approximately rank 1. Note that if $W^{(1)}$ is rank 1, $W^{(1)}X$ will also be rank 1.
>
> > Also, in the theoretical part the authors transition from gradient flow to discrete GD, introducing $\eta^2$ terms that appear later without clear transition.
>
> In Phase I, i.e. the progressive sharpening phase, the dynamic approximately follows the gradient flow trajectory. This has been empirically verified by [CKL+ 21] and assumed by [WLL22]. Hence, we analyze gradient flow in Phase I for simplicity. However, for later phases, the gradient descent trajectory differs from that of gradient flow, especially during the sharpness reduction phase where the $\eta^2$ term plays a key role. Then it is not valid to focus on gradient flow and therefore we transition to GD and analyze its dynamics in detail.
>
> [CKL+ 21] Jeremy M. Cohen, Simran Kaur, Yuanzhi Li, J. Zico Kolter, and Ameet Talwalkar. Gradient descent on neural networks typically occurs at the edge of stability. In 9th International Conference on Learning Representations, ICLR 2021,
>
> [WLL22] Zixuan Wang, Zhouzi Li, and Jian Li. Analyzing sharpness along GD trajectory: Progressive sharpening and edge of stability. Advances in Neural Information Processing Systems, 35:9983–9994, 2022.
>
> >Minor: figures suffer from small font sizes and thin lines, which may hinder readability.
>
> > Minor: the individual eigenvector target alignment plots are on a linear scale, making it hard to assess contributions from low-alignment modes. A log-scale plot would provide a clearer picture of the alignment shift across all modes.
>
> > Minor: the sentence _Note that the alignment values sum up to 1, so they can be regarded as a kind of “distribution”_ is a bit informal a not necessary in the definition.
>
> We thank the reviewer for these helpful suggestions and will make edits in our paper.

---

> > ### Comment · Reviewer_GAQw · 2025-08-07
> > **Response**
> >
> > I thank the authors for their response. In particular, I appreciate the extra experiments on the learning rate grid and the rank of the weights. I hope they will incorporate in the main text.
> >
> > On the rank of the weight matrices: there is a line of work showing what happens at the rank of the weights during training (e.g. Martin and Mahoney, https://www.jmlr.org/papers/volume22/20-410/20-410.pdf). In particular, the rank 1 assumption cannot happen at time 0 for gaussian random matrices under the standard random matrix theory scaling (which is also used by standard neural network initialization schemes). Hence, this concern remains.
> >
> > I increase my score to 4 because the other concerns are adequately addressed.

---

> ### Author Response · Authors · 2025-08-08
> **More explanations on the rank 1 assumption**
>
> Thanks a lot for your comment and for raising the score. We would like to elaborate more on the rank 1 structure here. We want to clarify that our rank 1 assumption is not made at time 0, but at some $t_0$ in the early training period, as stated in Assumption 4.1. We clearly know that due to Gaussian initialization, the initial weight matrices cannot have low-rank structures. However, the weights during training have the tendency to become of low rank, for example, the decreasing trend of the soft rank mentioned by Martin and Mahoney. The new experiments we added in our rebuttal also reveal this trend. Initially, the top 6 eigenvalues of $W^{(1)}$ are [3.9297, 3.8144, 3.8057, 3.7141, 3.6817, 3.6167] where there is no low-rank structure. However, a low-rank trend occurs as we train the model. (see more details in our rebuttal above to your questions)
>
>  In the two-layer linear network considered in our theoretical analysis, we can prove that $W^{(1)}$ becomes an approximately rank-1 matrix after an early training period. The main intuition is that during the early training period, the network output is small, and thus $W^{(1)}_{t+1}-W^{(1)}_t=-\frac{\eta}{n}W_t^{(2)T}(W^{(2)}_tW^{(1)}_tX-Y)X^T\approx\frac{\eta}{n}W^{(2)T}YX^T$ and $W^{(2)}\_{t+1}-W^{(2)}_t=-\frac{\eta}{n}(W^{(2)}_tW^{(1)}_tX-Y)X^TW_t^{(1)}\approx\frac{\eta}{n}YX^TW_t^{(1)}$. We can see that: i) the update of $W^{(1)}$ is approximately linear in $W^{(2)}$ and the update of $W^{(2)}$ is approximately linear in $W^{(1)}$; ii) since we consider the 1-dim output case, $W^{(2)}$ and $Y$ are both row vectors, and thus the update of $W^{(1)}$ is of rank 1. Given the above two facts, we can solve the approximate dynamics and get that $W^{(1)}$ will exhibit a rank-1 structure $W\_{t_0}^{(1)}\approx u\_{t_0}z\_{t_0}^T$ after some early time $t_0$, where $u\_{t_0}$ approximately aligns with $W^{(2)}\_{t_0}$. This low-rank trend in the early training period has also been studied in prior works, e.g. [1] and [2]. Finally, we want to emphasize again that we only make this rank 1 assumption at time $t_0$. If this holds at any time $t_0$, we prove instead of assuming that the structure will be preserved for $t>t_0$. (Appendix B.2).
>
>
> [1] Alexander B. Atanasov, Blake Bordelon, and Cengiz Pehlevan. Neural networks as kernel learners: The silent alignment effect. In The Tenth International Conference on Learning Representations, ICLR 2022.
>
> [2] Kaiqi Jiang, Dhruv Malik, and Yuanzhi Li. How does adaptive optimization impact local neural network geometry? Advances in Neural Information Processing Systems, 36, 2024

---

### Official Review · Reviewer_chUD · 2025-07-02

**Clarity:** 2
**Significance:** 2
**Originality:** 2
**Rating:** 4
**Confidence:** 4

**Summary:**

This paper investigates the evolution of the Neural Tangent Kernel (NTK) during gradient descent (GD) training, particularly focusing on the alignment between NTK eigenvectors and the target output.
It highlights a phenomenon termed "alignment shift," that higher learning rates induce stronger alignment of leading NTK eigenvectors with the target labels.
The authors validate this empirically across multiple architectures (FCN, ViT,) and support it with theoretical analysis on two-layer linear networks.

**Questions:**

Why the loss spikes always co-occur with the sharpness reduction? any intuition here?

**Ethical Concerns:**

["NO or VERY MINOR ethics concerns only"]

**Final Justification:**

Bordline paper. Given the insight that sharpness reduction induces kernel target alignment, I am lean to acceptance.

**Limitations:**

1. The experiments are limited to the binary classification.
2. The theoretical analysis relies on many assumptions with most assumptions being un-validated.

**Quality:**

2

**Strengths And Weaknesses:**

[Strength]
1. The authors study a quite under-explored problem: how the eigenvectors of NTK rotate during the GD training process, and how does these connect to the EoS phenomenon.
2. The authors analyzes the phenomenon from a theoretical analysis on a two-layer linear network. Their theory strengthen their empirical findings.

[Weakness]
1. The paper’s main empirical observation, that higher learning rates lead to stronger alignment of leading NTK eigenvectors with the target, is somewhat intuitive.
   - Larger learning rate (within a reasonable range) usually leads to better generalization, thus indicating better alignment between NTK and the target. The NTK at the local iterate somewhat represents the neural network in function space.
   - Better alignment between NTK and the target indicates better alignment between leading eigenvector of NTK with the target.
2. The theoretical analysis largely follows WLL22. Both study the NTK evolution of a two-layer linear network and approximate the NTK evolution with the norm of the second-layer weights. Novelty of the theory should be hightlighted.
3. It is unclear that if the increase of KTA always associated with the sharpness reduction. For example, in fig 3 (c), during the sharpness reduction phase, KTA first decrease and then increases. Also, the experiments of Central Flow cannot fully support this claim. Central flow is a good approximation of GD. GD generalizes better than GF. Thus, KTA of Central flow should be larger than GF.
4. Clearly, more experiments on more general settings are needed. The datasets are limited to binary classification, and more LRs should be tested. What if we consider the multi-class classifiction? and what if consider larger/smaller LRs?

---

> ### Author Rebuttal · Authors · 2025-07-30
>
> We thank the reviewer for their comments and would like to address their concerns below.
>
> > The paper’s main empirical observation, that higher learning rates lead to stronger alignment of leading NTK eigenvectors with the target, is somewhat intuitive.
>
> We agree that our main observation is somewhat intuitive, but we would argue that it is far from obvious. Although it is known that a larger learning rate (within a reasonable range) usually leads to better generalization, the underlying mechanism is not fully understood yet, and the connection between the generalization ability and the kernel target alignment is not obvious either. The traditional viewpoint focuses on the curvature perspective and argues that large learning rates find “flatter minima”, which generalize better. Our work supports an alternative perspective on the beneficial effects of large learning rates, in that they enhance feature learning by improving the alignment between the NTK and the target vector. We also study the connection between our findings and the generalization ability in lines 147-156 and Appendix A.5 in detail. We think such studies are valuable since the connections are not that obvious.
>
> > The theoretical analysis largely follows WLL22. Both study the NTK evolution of a two-layer linear network and approximate the NTK evolution with the norm of the second-layer weights. Novelty of the theory should be highlighted.
>
> A key innovation of this work, relative to prior works, is to demonstrate the role of the $O(\eta^2)$ term in contributing to the alignment shift (as noted by reviewer GAQw).  This term was neglected in WLL22 and other prior works, but is necessary for understanding the alignment shift phenomenon that we study.
>
> > It is unclear that if the increase of KTA always associated with the sharpness reduction. For example, in fig 3 (c), during the sharpness reduction phase, KTA first decrease and then increases.
>
> Our claim is not that each step during the sharpness reduction phase will cause the KTA to rise, but rather that the overall phase will cause the KTA to rise (relative to its behavior under the gradient flow).  Note that in this figure, we computed the KTA every 5 steps.  If we had instead computed the KTA at every step, then you would see period-2 oscillations in the KTA.  These are intuitively caused by the sign-flipping oscillations in $\langle E_t, q_1 \rangle$, where $E_t$ is the error vector and $q_1$ is the top eigenvector of the data kernel. (See the theoretical analysis for more detailed intuition.) The important story in Figure 3 is that over the whole course of a sharpness reduction phase, the KTA increases in a 'stairwise' fashion after each phase.
>
> Suppose we really want to study the connection between the evolution of KTA and sharpness at each step. In that case, we will need to focus on the time-averaged trajectory to get rid of the influence of the oscillation. That's why we introduce the experiments under Central Flow.  In Figure 4(b), we 'branch off' gradient flow from the central flow at three different points in training, and observe that the KTA of the gradient flow is lower than the KTA of the central flow at each step after branching off.
>
>
> > Also, the experiments of Central Flow cannot fully support this claim. Central flow is a good approximation of GD. GD generalizes better than GF. Thus, KTA of Central flow should be larger than GF.
>
> As mentioned above, our central flow experiments show that KTA under central flow is larger than that under GF. To see why this can support our claim, we want to highlight that Central Flow is a **sharpness penalized** gradient flow.  Therefore, our central flow experiments demonstrate that adding a sharpness penalty to the gradient flow promotes the increase of KTA, thus highlighting the connection between sharpness and KTA evolution from another perspective.
>
> > Clearly, more experiments on more general settings are needed. The datasets are limited to binary classification, and more LRs should be tested. What if we consider the multi-class classifiction? and what if consider larger/smaller LRs?
>
> We would like to clarify that we do include more experiments in more settings in the appendices. The datasets include image tasks (in the main part of the paper and Appendix A.2.1 and A.2.2) and language tasks (Appendix A.2.3), and multi-class classification in Appendix A.2.4. For more experiments with more LRs, please see the first point of the responses to Reviewer GAQw.
>
> > Why the loss spikes always co-occur with the sharpness reduction? any intuition here?
>
> The main intuition and underlying mechanism are discussed in detail in Section 4.2, the four phases of the training dynamics. We would like to make a brief summary here. First, we show that the evolution of the approximate sharpness, i.e. the norm square of the second-layer weights, is determined by $-\langle E_t, F_t\rangle$, where $F_t$ is the network output and $E_t$ is the error vector. At first, $\langle E_t, F_t\rangle<0$, causing $-\langle E_t, F_t\rangle>0$ and the sharpness to increase (phase I). When the sharpness is large enough, it will cause $E_t$ to oscillate with increasing magnitude along the $q_1$ direction, where $q_1$ is the top eigenvector of the data kernel. (see Lemma 4.4) That will lead to the growth of $\Vert E_t\Vert$. When $\Vert E_t\Vert$ grows large enough, we will see the loss spike, and in the meantime, $\langle E_t, F_t\rangle$ will become positive, i.e. $-\langle E_t, F_t\rangle<0$, making the sharpness decrease. That's why the loss spike is usually co-occur with the sharpness reduction.
>
> > The theoretical analysis relies on many assumptions with most assumptions being un-validated.
>
> For more explanations on the intuition and validation of our assumptions, please see the first point of the response to Reviewer fnJU.

---

> > ### Comment · Reviewer_chUD · 2025-08-05
> >
> > Thank you for resolving some of my misunderstandings. I will raise my score from 3 to 4.

---

> > > ### Author Response · Authors · 2025-08-05
> > >
> > > Thanks a lot for your comment and for raising the score!

---

### Official Review · Reviewer_6a2Z · 2025-07-09

**Clarity:** 4
**Significance:** 3
**Originality:** 4
**Rating:** 6
**Confidence:** 3

**Summary:**

This article takes a novel attempt to describe NTK eigenvector behavior during the Edge of Stability phase. They observe that larger learning rates cause the top eigenvectors to align more closely with the target during EoS, coinciding with a decrease in model sharpness, as supported by empirical observations. The authors support these findings with a theoretical analysis of a linear network trained with gradient flow on MSE and further analyze the relationship between sharpness reduction and kernel alignment through the central flow framework.

**Questions:**

- Figures 1(c) (bottom) and 3(c) (bottom): The curves do not appear to overlap with their respective counterparts (lr = 0.1 & 0.01) from 1(a) and 3(a). Is this alignment between the top NTK eigenvector and Y instead (as in 2(c))?
- Why is the Individual Eigenvector Target Alignment vs. Eigenvector index peak in Figs. 1 - 3 (b) close to but $\neq0$?

**Ethical Concerns:**

["NO or VERY MINOR ethics concerns only"]

**Limitations:**

Yes.

**Paper Formatting Concerns:**

N.A.

**Quality:**

4

**Strengths And Weaknesses:**

**Strengths**
- The article provides a clear, cohesive, and detailed approach to studying the behavior of the NTK eigenvector during the Edge of Stability phase in DNNs. This serves as a first-of-its-kind attempt to investigate this property, contrary to existing literature, which has focused on deciphering the reasons behind and the nature of the eigenvalue spectra instead
- The authors conduct experiments over a) 4-Layer GeLU networks trained on (a binary class subset of) CIFAR-10, b) 2-layer linear models, c) vision transformer setups to provide empirical evidence for the argument that increasing learning rate causes
  - An increase in overall NT-kernel alignment with the target $\to$ indicating better generalization
  - A shift of the individual NTK-eigenvector -- target alignment towards the top eigenvectors of the NTK $\to$ supporting the point that there is an increased bias of the model on the leading eigenvector directions of the NTK (which capture the signal(s))
  - Sudden drops in the sharpness over training coincide with a sharp rise in the KTA
- Leveraging established connections between the Hessians' spectra and the (norm-squared of the) 2nd layer weight matrix of the 2-layer linear model under consideration, the authors perform a careful study of the evolution of the latter in terms of arriving at the EoS phase. Through this analysis, they demonstrate that a decrease in this proxy object (around the stability threshold) coincides with a shift in the alignment weights towards the top eigenvectors of the NTK

---

> ### Author Rebuttal · Authors · 2025-07-30
>
> We thank the reviewer for their comments and would like to answer their questions below.
>
> > Figures 1(c) (bottom) and 3(c) (bottom): The curves do not appear to overlap with their respective counterparts (lr = 0.1 & 0.01) from 1(a) and 3(a). Is this alignment between the top NTK eigenvector and Y instead (as in 2(c))?
>
> Figures 1(c) bottom and 3(c) bottom are both about kernel target alignment, same as in 1(a) and 3(a). The curves do not overlap mainly because in 1(a) and 3(a), we plot the curves every 50 and 20 iterations respectively in a long time range to get an overall behavior of the evolution of KTA, while in 1(c) bottom and 3(c) bottom, we zoom in the EoS period and plot the curves every 3 and 5 steps respectively to study the detailed connection between the evolution of KTA and that of the sharpness.
>
> > Why is the Individual Eigenvector Target Alignment vs. Eigenvector index peak in Figs. 1 - 3 (b) close to but $\neq0$?
>
> This is somewhat common behavior, and we don't have a complete understanding why.   Note that in these situations, the NTK's eigenvector with index 0 usually corresponds to the all 1's vector, i.e. the "DC component."   This is true both before and after feature learning.  Since it is always the all 1's vector, it cannot align with the target.

---

### Decision · Program_Chairs · 2025-09-17

**Decision:**

Accept (poster)

**Comment:**

This paper theoretically and empirically investigate an important problem how the NTK evolves during optimization at the edge of stability. It is shown that the top eigenvector of the NTK aligns more to the feature direction with a larger step size, and in such a setting, the model's sharpness decreases. The theoretical analysis was conducted on a two layer linear model.

The paper is well written. The NTK dynamics was carefully investigated through a detailed and comprehensive numerical experiments with several models. The theoretical analysis is performed only on a linear network but it gives a precise description of the dynamics. Then, the paper successfully resolved the problem that has not been well addressed in previous work.

I think this paper is beneficial to the community, hence recommend acceptance.